# 3D single cell migration driven by temporal correlation between oscillating force dipoles

**Amélie Luise Godeau[1†], Marco Leoni[2,3†], Jordi Comelles[1], Tristan Guyomar[1], Michele Lieb[1], Hélène Delanoë-Ayari[4], Albrecht Ott[5], Sebastien Harlepp[6], Pierre Sens[2]\*, Daniel Riveline[1]\***

[1]Laboratory of Cell Physics, ISIS/IGBMC, UMR 7104, Inserm, and University of Strasbourg, Strasbourg, France; [2]Institut Curie, Université PSL, Sorbonne Université, CNRS UMR168, Laboratoire Physico Chimie Curie, Paris, France; [3]Université Paris-Saclay, CNRS, Laboratoire de l'accélérateur linéaire, Orsay, France; [4]Univ. Lyon, Université Claude Bernard Lyon 1, CNRS, Villeurbanne, France; [5]Saarland University, Center for Biophysics, Biologische Experimentalphysik, Saarbrücken, Germany; [6]Tumor Biomechanics, INSERM UMR S1109, Institut d'Hématologie et d'Immunologie, Strasbourg, France

**\*For correspondence:**
pierre.sens@curie.fr (PS);
riveline@unistra.fr (DR)

[†]These authors contributed equally to this work

**Abstract** Directional cell locomotion requires symmetry breaking between the front and rear of the cell. In some cells, symmetry breaking manifests itself in a directional flow of actin from the front to the rear of the cell. Many cells, especially in physiological 3D matrices, do not show such coherent actin dynamics and present seemingly competing protrusion/retraction dynamics at their front and back. How symmetry breaking manifests itself for such cells is therefore elusive. We take inspiration from the scallop theorem proposed by Purcell for micro-swimmers in Newtonian fluids: self-propelled objects undergoing persistent motion at low Reynolds number must follow a cycle of shape changes that breaks temporal symmetry. We report similar observations for cells crawling in 3D. We quantified cell motion using a combination of 3D live cell imaging, visualization of the matrix displacement, and a minimal model with multipolar expansion. We show that our cells embedded in a 3D matrix form myosin-driven force dipoles at both sides of the nucleus, that locally and periodically pinch the matrix. The existence of a phase shift between the two dipoles is required for directed cell motion which manifests itself as cycles with finite area in the dipole-quadrupole diagram, a formal equivalence to the Purcell cycle. We confirm this mechanism by triggering local dipolar contractions with a laser. This leads to directed motion. Our study reveals that these cells control their motility by synchronizing dipolar forces distributed at front and back. This result opens new strategies to externally control cell motion as well as for the design of micro-crawlers.

## Editor's evaluation

This manuscript suggests a novel mechanism by which an animal cell can move through a three-dimensional extracellular matrix, namely by synchronized oscillations in contraction at the front and at the back of the cell. It should be of great interest to a variety of researchers in cellular biophysics.

## Introduction

Cell motility is essential in a variety of biological phenomena (*Yamada and Sixt, 2019*). Cells move during development in the presence or in the absence of chemical gradients. Their relevant localizations at the

right moment is essential to secure completion of viable developing embryos. Also, defects in migrations are known to be involved in cancer progression with the increased motility of cells during invasion (*Stuelten et al., 2018*). Understanding cell motion in physiological environment is therefore recognized as a central question in basic and in applied sciences.

In this context, the determination of genetic and proteins networks has been instrumental in identifying the central pathways at play for motility (*Jia et al., 2022*). These steps in outlining specific partners are important in localizing defects and they open potential perspectives for testing new strategies to act on these situations where cell migration is impaired in vivo and in vitro. However, cell motion calls also for physical principles because motion at this scale is not intuitive (*Tanimoto and Sano, 2014*). This requires alternative approaches in synergy between experiments and theory.

Cells motion in 2D flat surfaces has been documented in details (*Alberts, 2002*). The classical image of extension of lamellipodia and retraction of the back of the cells is known. Cells crawl and protrusions grow and retract around the cell. New dynamics are also associated with the motion of confined cells in microfabricated channels where retrograde flow of actin at the scale of the entire cell was reported to play central roles in motility (*Liu et al., 2015*; *Reversat et al., 2020*). However, both approaches – cells crawling on 2D flat surfaces and in channels – call for tests in actual 3D situations where cells interact with an environment similar to physiological 3D situations. This experimental challenge of mimicking in vivo environments goes together with theoretical basic questions.

At cellular scales inertia is negligible which puts some particular constraints on self-propelled microscopic objects (*Purcell, 1977*; *Tanimoto and Sano, 2014*). Low Reynolds number micro-swimmers in Newtonian fluids must obey the scallop theorem, stating that the cyclic sequence of shape changes they perform to swim cannot be symmetrical in time (*Purcell, 1977*; *Najafi and Golestanian, 2004*; *Golestanian and Ajdari, 2008*; *Barry and Bretscher, 2010*; *Leoni and Sens, 2015*). In his seminal work, Purcell illustrated such behavior as closed trajectories encompassing a finite area in some properly chosen phase space (*Purcell, 1977*).

Crawling cells are not strictly bound to obey the scallop theorem, which stems from the time reversibility of the Stokes equation. Cells moving in 2D or in micro-channels often display clear spatial polarization, characterized by F-actin flowing from the front to the back of the cell (*Barnhart et al., 2011*; *Liu et al., 2015*). For such fast moving cells, the lack of time reversal symmetry – the fact that the system looks different when the movie is played backward – clearly manifests itself by the existence of a coherent actin flow at the scale of the entire cell and the cell velocity is directly related to the dynamics of the actin flow. This mode of motility has been studied in detail theoretically, in particular to understand how the positive feedback between actin flow and the distribution of contractile units (myosin motors) may lead to spontaneous symmetry breaking (*Blanch-Mercader and Casademunt, 2013*; *Recho et al., 2013*; *Recho et al., 2014*; *Maiuri et al., 2015*). Many cells, especially mesenchymal cells, do not show such polarized actin flow at the scale of the entire cell, but instead display cycles of protrusion/retraction at both ends of the cell. The actin dynamics within protrusions is comparable to that of fast-moving cells, but the existence of multiple competing protrusion leads to slow cell translocation (see for example *Caballero et al., 2014*; *Lo Vecchio et al., 2020*). Slow moving cells in complex environments such as the extracellular matrix (ECM) could leverage the visco-elastic nature of the environment (*Qiu et al., 2014*; *Datt et al., 2018*) or the complex dynamics of adhesion and detachment (*Wagner and Lauga, 2013*; *Leoni and Sens, 2017*). This makes the search for unifying principles underlying 3D cell movement challenging.

In this article, we report the experimental design of a 3D physiological matrix which single cells deform while moving. We label fluorescently the important cellular proteins involved in cell motility. Deformation of the matrix correlated with motility/adhesion proteins localizations allow us to show that two forces dipoles mediated by acto-myosin at each side of the nucleus are correlated with a phase shift when cells move directionally. We propose a model which is consistent with this explanation and test this mechanism by inducing force dipoles. Our results shed light on generic physical mechanisms at play in cell migration.

## Results

### Fibroblasts in CDMs generate contractile-extensile regions on either side of the nucleus

To search for generic readouts for 3D cell migration, we designed an assay where cell migration and the spatial distribution of cell-matrix interactions could be tracked simultaneously and quantified over time.

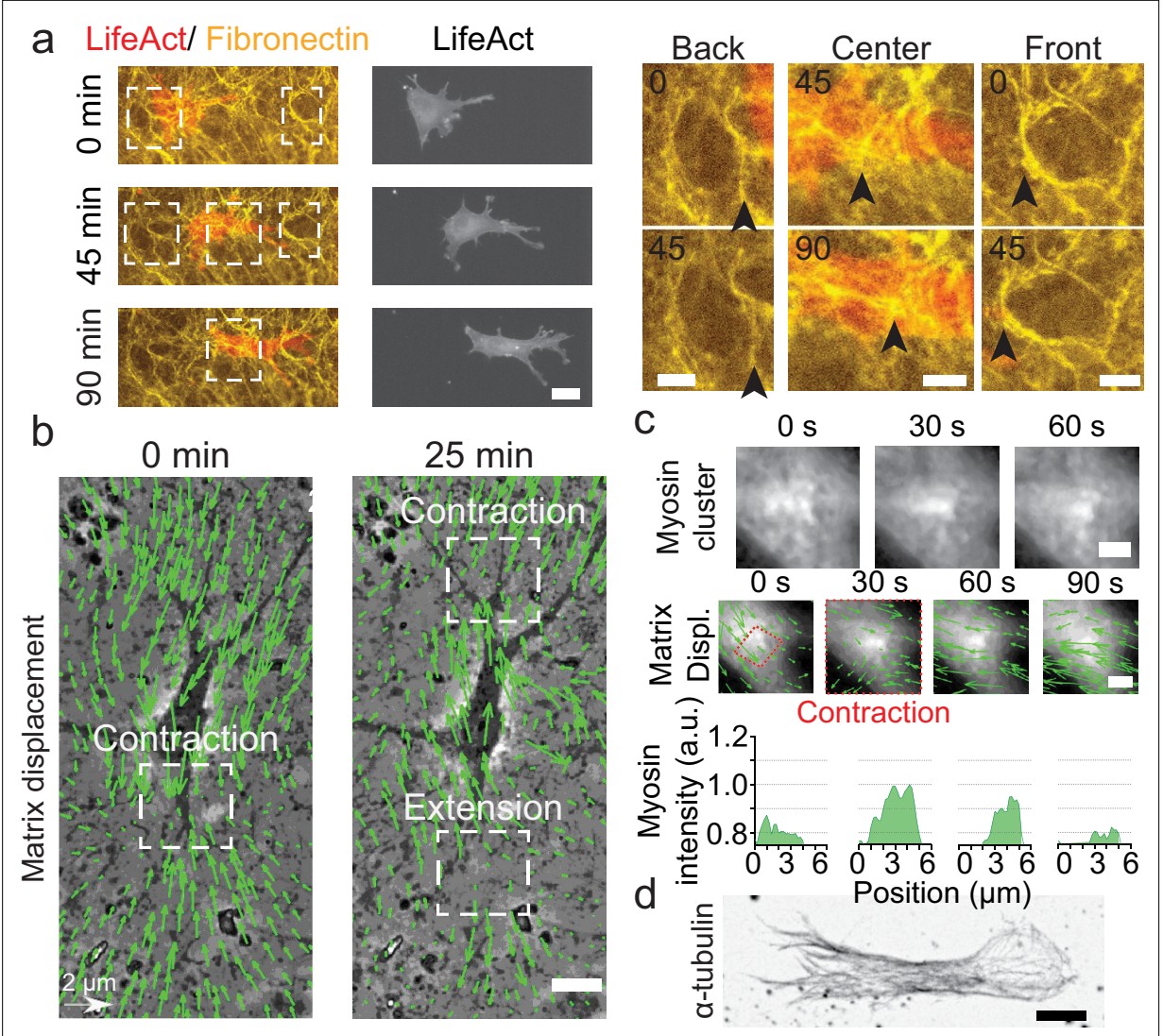

**Figure 1.** Key players in cell motility. (**a**) Left panel (and *Video 1*): A cell deforms the fibronectin (FN) network when migrating (FN in yellow and mCherry-LifeAct for actin filaments in red). Right panel: Enlargement of the white windows of the left panel. Black arrows highlight displacement of fibers due to cell movement. (**b**) Overlay of phase contrast image and Kanade-Lucas-Tomasi (KLT) calculation of mesh displacement (green arrows – with scale bar shown, arrows indicate displacement between two consecutive frames, $\Delta t$=5 min) with local contraction and extension regions indicated with white windows. (**c**) Myosin clusters (upper panels) form locally within cells and are correlated with local contraction: KLT deformation (green arrows, arrows indicate displacement between two consecutive frames, $\Delta t$=30 s) and myosin-mCherry signal (middle panels). Average myosin intensity profile along the red dashed square of the frames shown in the upper panels (lower panels). (**d**) α-Tubulin staining of a cell inside cell derived matrix (CDM). Microtubules extend from the centrosome to the periphery of the cell (see also *Video 4*). Scale bars: (**a**) 25 µm (10 µm in the enlargements); (**b,c,d**) 10 µm.

The online version of this article includes the following source data and figure supplement(s) for figure 1:

**Source data 1.** CDM characterization.

**Figure supplement 1.** Characterization of cell derived matrix (CDM).

**Figure supplement 2.** Cells deform the cell derived matrix (CDM) network.

**Figure supplement 3.** Different types of fibroblasts generate contractile-extensile regions on either side of the nucleus in cell derived matrices (CDMs).

In our experiment, NIH 3T3 fibroblasts moved inside a fluorescently labelled cell derived matrix (CDM) (*Figure 1a*, left panels and *Video 1*), obtained using a protocol adapted from (*Cukierman et al., 2001*). Briefly, a confluent monolayer of NIH3T3 cells expressing fluorescently labeled fibronectin was triggered to synthesize ECM proteins for 8 days. Then, cells were removed by lysis, generating a 3D protein meshwork (*Figure 1—figure supplement 1a*). The resulting CDM had large pores inside, but it was covered by a

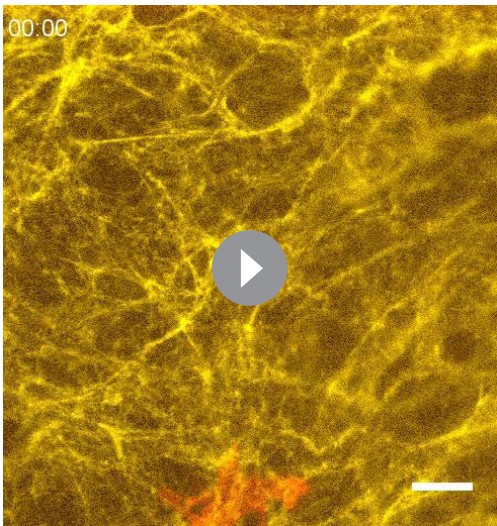

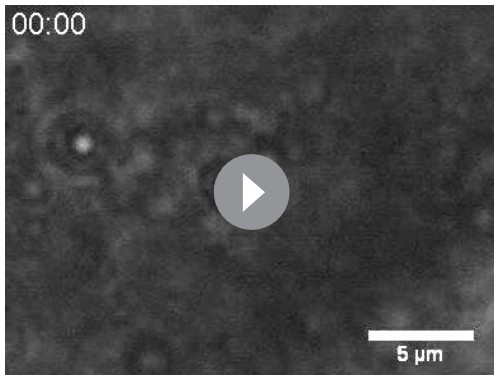

**Video 1.** NIH3T3 fibroblast transfected with mCherry-LifeAct deforms the fibronectin network in yellow while moving, scale bar 20 µm, time in hh:mm.
https://elifesciences.org/articles/71032/figures#video1

**Video 2.** Cell derived matrix (CDM) is elastic, as shown by optical tweezer characterization, time in mm:ss.
https://elifesciences.org/articles/71032/figures#video2

dense crust, as shown by electron microscopy (EM) (*Figure 1—figure supplement 1b*). Despite this, 3T3 fibroblasts seeded on top of the CDM passed through and spread within the matrix, which was significantly thicker than cells themselves (*Figure 1—figure supplement 1c and d*). We then determined by micromanipulation of embedded beads the mechanical properties of these cell derived matrices (*Figure 1—figure supplement 1e*, *Video 2*, and Appendix 1). Optical tweezer experiments showed no hysteresis in the displacement curves, suggesting that the matrix behaved as an elastic material within the amplitudes and timescales explored (*Figure 1—figure supplement 1f and g*), which is in agreement with previous reports (*Petrie et al., 2012*). Moreover, the relationship between force and displacement was linear (*Figure 1—figure supplement 1h*) and we could determine the CDM's elastic modulus to be ≈50 Pa (*Figure 1—figure supplement 1h and i*) and no modification of the meshwork was observed after 24 hr of cell migration inside (*Figure 1—figure supplement 1j*). Altogether, these physiological 3D porous matrices were thick enough to host 3D cell migration and were sufficiently soft and elastic to relate cell dynamics to matrix deformation.

For migration experiments, 3T3 fibroblasts were plated at low density, spontaneously penetrated the matrix and could be followed individually. Cells displayed an elongated morphology (*Figure 1—figure supplement 2*), typical of fibroblasts in these soft and elastic matrices (*Cukierman et al., 2001*; *Petrie et al., 2012*; *Caballero et al., 2017*). We observed (i) a cortical distribution of acto-myosin (*Figure 1a*), (ii) the presence of focal adhesions distributed all over the cell membrane and mediating its adhesion to the surrounding matrix (*Figure 1—figure supplement 2* and *Video 3*), and (iii) the microtubule cytoskeleton expanding from the centrosome to the periphery of the cell (*Figure 1d* and *Video 4*). Strikingly, the CDM was easily deformed by cells as they moved through (*Figure 1a*, right panels and *Video 5*). This enabled us to quantify the associated matrix deformation (*Delanoë-Ayari et al., 2008*) via the Kanade-Lucas-Tomasi (KLT) tracker method (*Lucas and Kanade, 1981*; *Figure 1b*). These deformations were present at the front and at the back of the nucleus of polarized

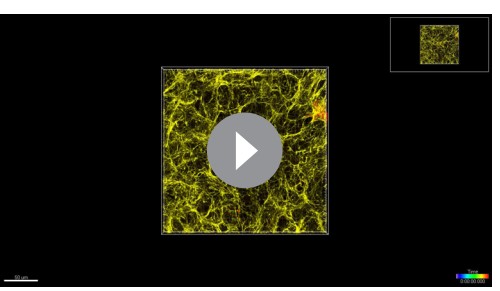

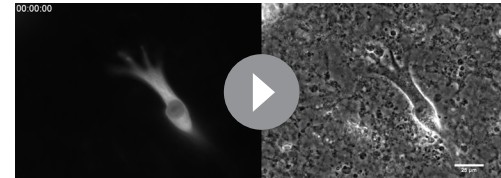

**Video 3.** Cells motion in 3D with cell derived matrix (CDM) and the associated focal contacts dynamics fibronectin in yellow and zyxin in red, time in hh:mm:ss.
https://elifesciences.org/articles/71032/figures#video3

**Video 4.** Microtubule asymmetric distribution (left) is associated with cell polarity during motion, time in hh:mm.
https://elifesciences.org/articles/71032/figures#video4

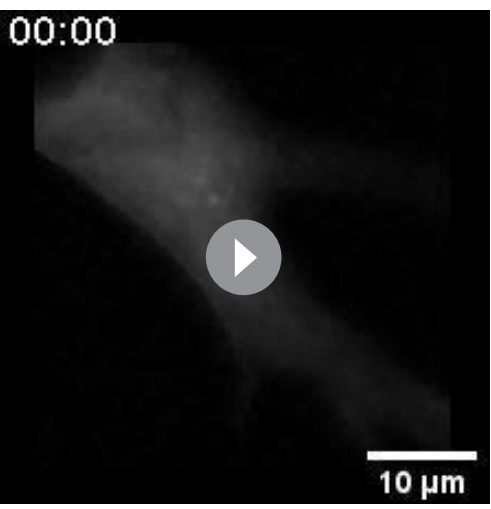

**Video 6.** Formation of myosin clusters simultaneously to contraction, time in mm:ss.

https://elifesciences.org/articles/71032/figures#video6

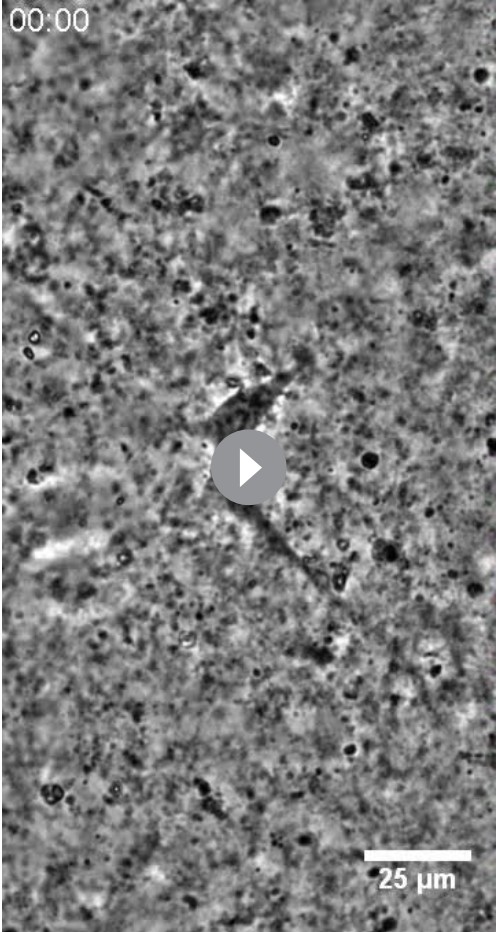

**Video 5.** Cell deforms the cell derived matrix (CDM), time in hh:mm.

https://elifesciences.org/articles/71032/figures#video5

cells, where phases with both contractile and extensile patterns were observed. Similar patterns were also observed in other cell types migrating in CDMs. Both mouse embryonic fibroblasts (MEFs) and rat embryo Fbroblasts 52 (REF52) exhibited phases of contraction and extension patterns at the front and the back of the nucleus (*Figure 1—figure supplement 3*). Moreover, the matrix deformation could be imaged concomitantly with the cellular machinery responsible for force generation. We could observe a transient densification of the myosin signal forming clusters of myosin associated with the contraction observed in the matrix displacement (*Figure 1c* and *Video 6*). This force transmission between the cell and the CDM was also observed at the focal contact level: enlargement of pores of the fibronectin meshwork suggested a local pulling force by the cells (*Figure 1—figure supplement 2*). Therefore, cells embedded in CDMs generate forces which translate into contractile and extensile deformations in the surrounding matrix.

## Dynamics of matrix deformation for migrating and non-migrating cells

During migration experiments, some cells showed persistent motion (*Video 7*) while others moved back and forth along a constant central position. Hereafter, we denote as migrating cells the first type and as non-migrating cells the second type. Interestingly, both migrating and non-migrating cells exhibited local zones of contraction-extension along time (*Figure 2a*). Such regions can be seen on both sides of the nucleus (*Figure 1b*). To characterize the behavior of these contraction-extension regions over time, we defined the cell axis along the direction of polarization (defined by a more protrusive cell end) for non-migrating cells and along the direction of migration for migrating cells (*Figure 2b and c*). Then, we projected the divergence of the matrix across the x-y plane on the cell axis for every timepoint and plotted the corresponding kymographs (*Figure 2d and e*). For non-migrating cells, we could observe sequences of positive and negative divergence along time (alternating blue and yellow area) on still regions (black lines in *Figure 2d*) at both sides of the nucleus. For migrating cells, sequences alternating positive and negative divergence were present as well (*Figure 2e*). However, these contraction-extension patterns were not still, but moved along with the cell (blue and red lines in *Figure 2e* correspond to the back and front of the nucleus, respectively). Thus, we suggest that these contraction-extension regions are equivalent to two force dipoles on either side of the nucleus.

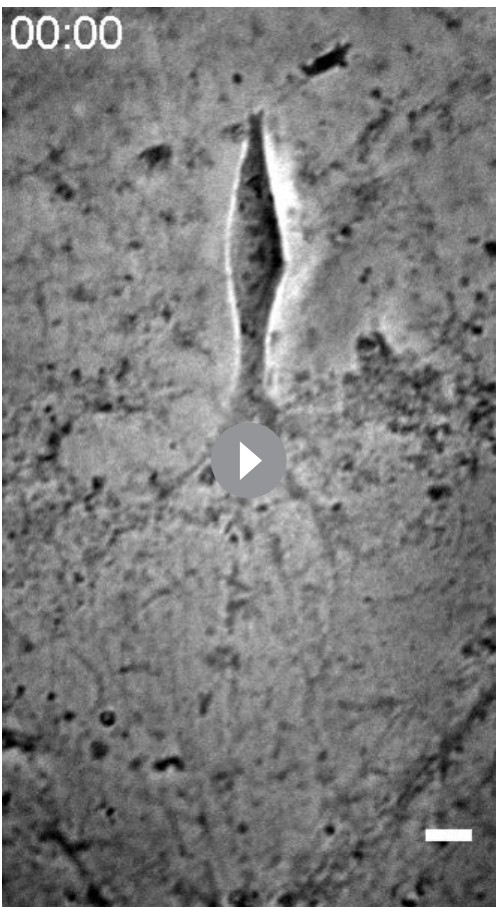

**Video 7.** Another example of cell motion in cell derived matrix (CDM), scale bar 25 μm, time in hh:mm. https://elifesciences.org/articles/71032/figures#video7

As it can be seen from the kymographs, these deformations were cyclic. To determine their period of contraction, we extracted temporal profiles of the divergence of the matrix displacement by averaging over the regions at the front and the back of the nucleus (*Figure 2—figure supplement 1a-d*). By doing so, we obtained the traces of the contractions at the front and the back over time for single cells (*Figure 2—figure supplement 1e*). Measurements of the magnitude of the peaks observed in the divergence traces showed no difference between front and back in migrating cells (*Figure 2—figure supplement 2*). The same behavior was observed for non-migrating cells, which were also similar in magnitude to the ones observed in migrating cells (*Figure 2—figure supplement 2*). We then performed the autocorrelation of these traces to obtain the period of the contractile and extensile patterns (*Figure 2—figure supplement 1f*). Autocorrelation of the divergence profile for individual non-migrating cells (*Figure 2f* – representative example for one cell) resulted in periods of oscillation similar at the front and back of the cell (≈6 min) (*Figure 2g*) (see Materials and methods section for quantification of the oscillations). For individual migrating cells, autocorrelation of divergence profiles (*Figure 2h* – representative example for one cell) gave oscillation periods of ≈8 min, similar at the front and the back as well (*Figure 2i*). These contraction relaxation cycles with periods of ≈8 min and amplitudes of typically 2.5 μm are reminiscent of previous reports of cell autonomous contractile oscillations (*Kruse and Riveline, 2011*). Periods of oscillations were comparable for motile and non-motile cells (*Figure 2g and i*). The phases of contraction seemed to correlate with the formation of local myosin clusters as described above (*Figure 1c*), which is consistent with the observation of distinct myosin-driven contraction centers in the migration of neurons on 2D surfaces (*Jiang et al., 2015*).

We observed more protrusive activity at one of the two ends of elongated cells, in both motile and non-motile cells. Since this apparent spatial polarization was not sufficient to elicit directed motion, we looked for other sources of symmetry breaking. We measured the temporal cross-correlation function of the contraction-extension cycles at the two cell ends (*Figure 2—figure supplement 1g* and *Figure 2j*). This revealed significant differences between migrating and non-migrating cells. While non-migrating cells systematically showed no phase shift between the two ends, with cross-correlation peaks localized at a time shift *t*=0, migrating cells showed cross-correlation peaks at non-zero time lags (*Figure 2k*), suggesting a phase shift between the front and back contraction-extension cycles (differences of phase shift in migrating and non-migrating cells were statistically significant with p=0.0246). A system undergoing directed motion must exhibit time reversal asymmetry. In our case, this asymmetry manifests itself through a time lag between the contraction-extension cycles of the two force dipoles.

## Multipole analysis of the matrix deformation

The most intuitive way to visualize time reversal symmetry breaking is through the existence of cycles in a properly chosen phase space. This phase space may be based on a multipole expansion of the traction force exerted by the cell on its surrounding. This approach was pioneered by Tanimoto and Sano for *Dictyostelium discoideum* crawling on deformable 2D substrate (*Tanimoto and Sano, 2014*). In particular, they showed that crawling *D. discoideum* exhibits cycles of finite area in the dipole-quadrupole phase space.

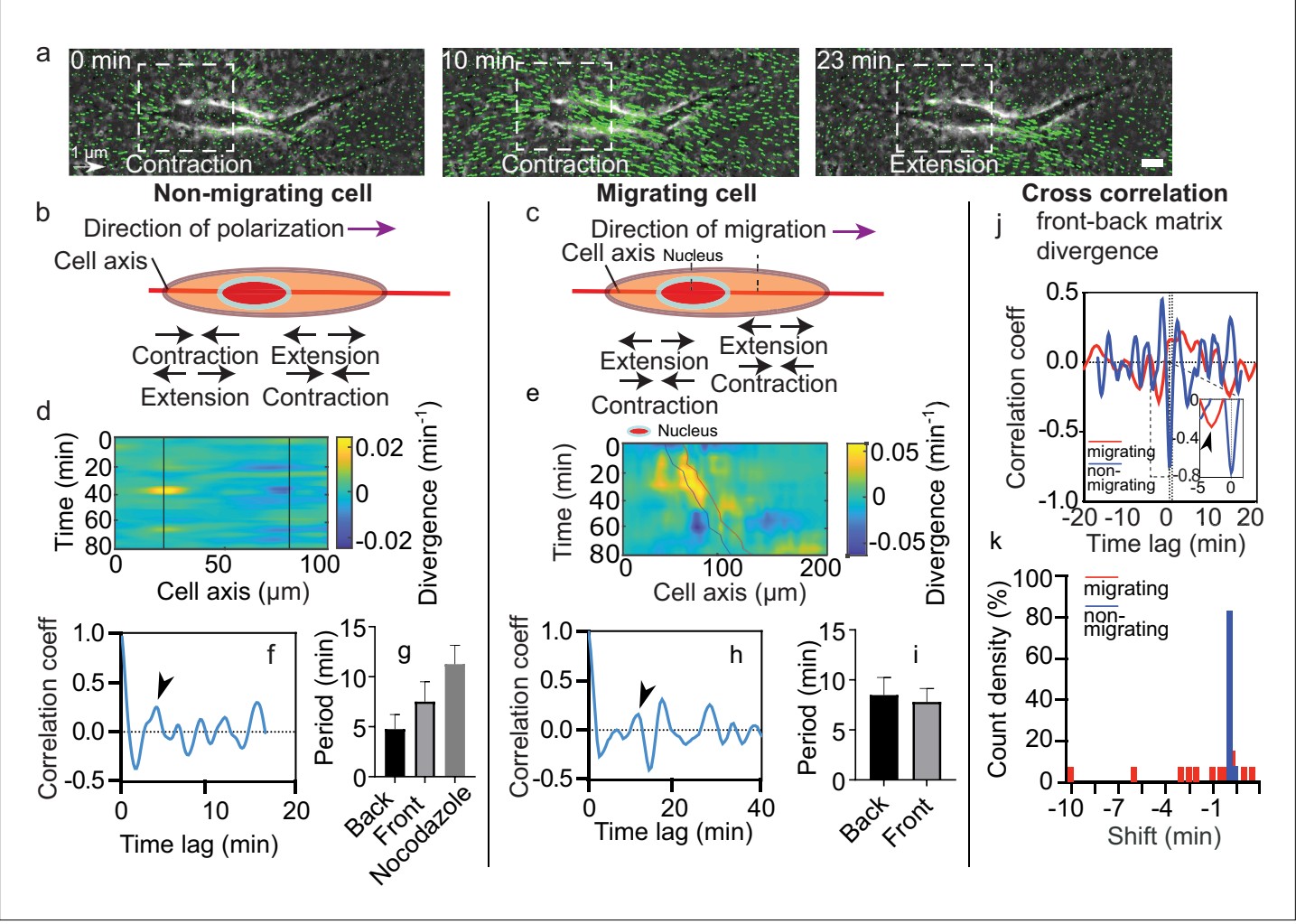

**Figure 2.** Dynamics of matrix deformation for migrating and non-migrating cells. (**a**) Snapshots overlaying phase contrast images and Kanade-Lucas-Tomasi (KLT) calculation of matrix rate of deformation (green arrows indicate displacement between two consecutive frames, Δt=1 min) showing a contraction/extension center, scale bar 10 µm, time in minutes. (**b–c**) Schematics of the alternating phases of contraction and extension for a non-migrating cell (**b**) and a migrating cell (**c**). (**d–e**) Heatmap of the divergence of the corresponding matrix displacement. Contractile and extensile force dipoles correspond to blue and yellow spots, respectively. Non-migrating cells (**d**) show two oscillating dipoles (their centers are approximately indicated by the solid black lines), while migrating cells (**e**) show a different spatio-temporal pattern with sequences of alternating positive and negative divergence. The blue and red solid lines in (**e**) indicate the two sides of the nucleus. (**f**) Correlation function of the contraction-extension time series at the back of a non-migrating cell, with a first peak at ≈5 min. Time trace 36 min. (**g**) Average periods of contraction-extension cycles: 4.8±1.5 min for the front, 7.5±2.0 min for the back, and 11.0±1.9 min for nocodazole-treated cells. Error bars represent standard error of the mean (SEM). (**h**) Correlation function of the divergence at the back of a migrating cell, with a first peak at ≈10 min. Time trace 100 min. (**i**) Average period of the contraction-extension cycles for migrating cells of 8.5±1.8 min for the front and 7.8±1.4 min for the back. (**j**) Typical cross-correlation function between back and front divergence for a migrating cell and a non-migrating cell. Time trace non-migrating cell 36 min and migrating cell 100 min. (**k**) Distribution of the values of the time lag for migrating and non-migrating cells. Distribution of time lag for migrating cells is statistically significant to the time lag for non-migrating cells (p=0.0246, Kruskal-Wallis test). Error bars represent SEM. t-Tests show differences in oscillation periods between motile, non-motile, and nocodazole-treated cells are not statistically significant (with $n_{mot}$ = 13 motile cells, $n_{nomot}$ = 6 non-motile cells, $n_{Noco}$ = 6, and N>3 biological repeats).

The online version of this article includes the following source data and figure supplement(s) for figure 2:

**Source data 1.** Matrix contraction.

**Figure supplement 1.** Analysis pipeline.

**Figure supplement 2.** Divergence amplitudes.

In the present work, the traction forces exerted by the cell are analyzed indirectly through the deformation of the matrix. Indeed, our characterization of the mechanical properties of the CDM showed that it locally behaves as a linear elastic material with a well-defined elastic modulus (*Figure 1—figure supplement 1f-j*). This supports the approximation of a linear relationship between the moments of the force distribution and the moments of the resulting matrix displacement distribution. Note however that the relationship could be more complex due to matrix heterogeneities.

We analyzed the 2D projection of the 3D deformation field of the CDM and computed the dipolar and quadrupolar moments of the rate of matrix deformation (see Appendix 2 and *Figure 3—figure supplement 1* for details). Briefly, we extracted the 2D projected velocity vector field $u_i^{(n)}$ for the component $i$ of the change of substrate deformation between two consecutive frames at position $n$ of the mesh. The dipole is a tensor defined as $S_{ij} = \frac{D_{ij} + D_{ji}}{2}$ with $D_{ij} = \sum_n \Delta_i^{(n)} u_j^{(n)}$ , where $\Delta_i^{(n)}$ is the $i$ th component of the vector joining the cell center and the point $n$ on the mesh. Similarly, the quadrupole tensor is defined as $Q_{ijk} = \sum_n \Delta_i^{(n)} \Delta_j^{(n)} u_k^{(n)}$ . The largest eigenvalues of the dipole and quadrupole tensors are defined as the main dipole, $D$, and the main quadrupole, $Q$, and the corresponding eigenvectors are defined as the main dipole and quadrupole axes. *Figure 3a* shows an example of the orientation of the main dipole, and *Figure 3b* shows a sketch of the physical/geometrical meaning of these quantities. The main dipole axis was aligned with the direction of motion – within experimental noise (see *Figure 3—figure supplement 2* for a quantification) – as was reported for 2D motion (*Tanimoto and Sano, 2014*).

We then determined the time variation of the main dipole and quadrupole. The level of traction exerted by the cell on the matrix observed for non-migrating cells is comparable in magnitude to those of migrating cells (*Figure 3—figure supplement 3b*, differences between migrating and non-migrating cells are not statistically significant with p=0.2766). This shows that the absence of migration is not due to the lack of traction forces (see also *Figure 3—figure supplement 1*). Identifying time reversal symmetry breaking phenomena is not straightforward in the temporal representation of *Figure 3c*. Therefore, we instead represent the cell trajectory in the dipole/quadrupole phase space, in which non-reversible periodic trajectories appear as closed cycles of finite area (*Tanimoto and Sano, 2014*). We indeed observe that trajectories showed a cycle enclosing a finite area for migrating cells, but a negligible area for non-migrating cells (*Figure 3d and e*, and more examples in *Figure 3—figure supplement 3*). To further test the relationship between finite cycle area and motility of individual cells, we analyzed the migration of a cell that underwent motion and then stopped (*Figure 3f–h* and *Video 8*). Strikingly, we observed that the cycles switched from a finite area while the cell was moving to a vanishing area when the cell stopped (*Figure 3h*). The finite area is a direct illustration of the fact that the phase shift between the contraction-extension cycles at the two ends of migrating cells observed in *Figure 2* also manifests itself in the pattern of matrix displacement. This is a clear signature of time reversal symmetry breaking. The quantification of the area enclosed by the trajectories in the D-Q plane of migrating and non-migrating cells is detailed in Appendix 2 and shown in *Figure 3—figure supplement 3c*. Both absolute areas (in µm⁵/min²) and normalized areas (in percent of the area of the rectangle fitting the trajectory) were substantially larger for migrating cells than for non-migrating cells (differences were statistically significant with p=0.0177 and p=0.0431, respectively). This supports our claim that the relevant difference between the two behaviors is the existence of a phase shift rather than a difference in the level of traction forces.

To test whether we could correlate the oscillations at the two cell ends with molecular actors, we tracked cell motion in the presence of the microtubule depolymerizing agent, nocodazole. Upon addition of the drug, cell migration stopped (*Figure 4—figure supplement 1h*). Some cells then underwent a forward backward movement in the matrix (*Video 9*), which hinted at an oscillation driven by local force dipoles. We observed a similar behavior in nucleus-free cells, generated when protrusions rupture from the cell. The resulting cytoplast underwent a forward backward movement in the CDM (*Figure 1—figure supplement 3c*, *Video 10*). They switched polarization over time, with a frequency of one direction reversal every ≈36 min. Although the nocodazole-treated cells did not maintain a fixed cell polarization either, we extracted matrix displacement which showed a typical period as for non-treated cells of ≈11 min (*Figure 2g*). Due to the lack or switch in polarity we could not define a back and a front. However, oscillations at different cell ends were in phase and cells did not show directed motion. This suggests that the coupling between oscillators needed to promote directed motion could involve microtubules. Indeed, the microtubular network spans the entirety of the cell body (*Figure 1d* and *Video 4*), and is thus a natural candidate to transfer information between the two ends of the cell. This suggestion is consistent with the notion that the microtubular network regulates the polarity axis of migrating cells (*Etienne-Manneville, 2004*; *Kaverina and Straube, 2011*).

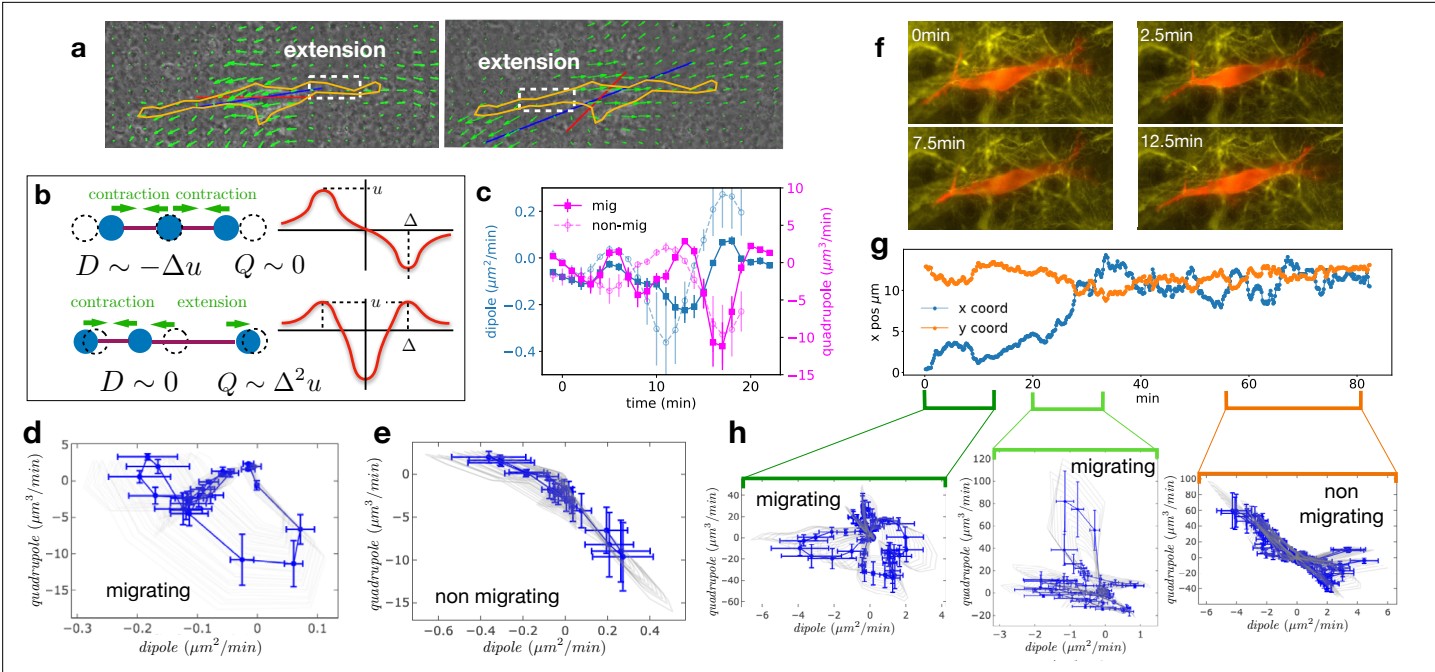

**Figure 3.** Multipole analysis of the matrix deformation rate. (**a**) Snapshots of a cell with: matrix rate of deformation, green arrows, the main dipole axis, blue, the axis of the cell motion, red. (**b**) Schematic representation of dipoles (**D**) and quadrupoles (**Q**) of the 1D matrix displacement (rate) distributions. The distribution on top has non-zero dipole but vanishing quadrupole, and that on the bottom has vanishing dipole and non-zero quadrupole. (**c**) Time series of the main dipole, blue, and quadrupole, magenta – projected on the cell axis – for a migrating cell (squares) and a non-migrating cell (circles), sampling approximately 1/10 of the duration of the entire experiment. (**d–e**) Cell trajectory in the dipole/quadrupole phase space for a migrating cell (**d**) and a non-migrating cell (**e**). The migrating cell follows a cycle with a finite area and the non-migrating cell does not. The error bars are obtained following the procedure described in Appendix 2. The individual cycles for different radii are shown in light gray. (**f**) Snapshots of a cell which in the course of the same experiment displays a transition from migrating to non-migrating behavior (LifeAct in red and fibronectin in yellow), scale bar 10 μm. (**g**) Cell positions in the x-y plane (blue and orange curves) showing a transition from migrating to non-migrating phase. (**h**) Trajectories in the dipole/quadrupole phase space for three different time intervals showing cycles with finite area in the migrating phase and with vanishing area in the non-migrating phase (see also ***Video 8***).

The online version of this article includes the following source data and figure supplement(s) for figure 3:

**Source data 1.** Dipole and quadrupole moments.

**Figure supplement 1.** Method to compute the multipolar terms.

**Figure supplement 2.** Quantification of cell and dipole orientation.

**Figure supplement 3.** Quantification of cell motion.

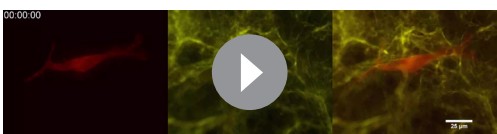

**Video 8.** Phase shift between local dipoles is associated with cell motion; cell motion (left, LifeAct red), fibronectin network deformation (center yellow), merge, time in hh:mm:ss.
https://elifesciences.org/articles/71032/figures#video8

## Persistent speed is related to the period of oscillations

The geometry and dynamics of the distribution of matrix displacement call for a direct comparison with models of self-propelled objects made of discrete moving beads (***Najafi and Golestanian, 2004***; ***Golestanian and Ajdari, 2008***; ***Wagner and Lauga, 2013***; ***Leoni and Sens, 2015***; ***Leoni and Sens, 2017***; ***Datt et al., 2018***). ***Figure 4a and b*** displays an idealized cell with two oscillating pairs of beads exerting time-shifted oscillatory force dipoles at its two ends. Here, the contractions at either ends of the cell are chosen to have the same magnitude for simplicity and in agreement with our observations (***Figure 2—figure supplements 1 and 2***), but this is not required to elicit net cell translocation. The cell activity is characterized by the amplitude $d$ and period $T$ of the oscillations and a phase shift $\psi = 2\pi\Delta T/T$ between oscillations at the two ends. The simplest self-propelled object is a micro-swimmer embedded in a Newtonian fluid and migrating due to hydrodynamic interactions (***Najafi***

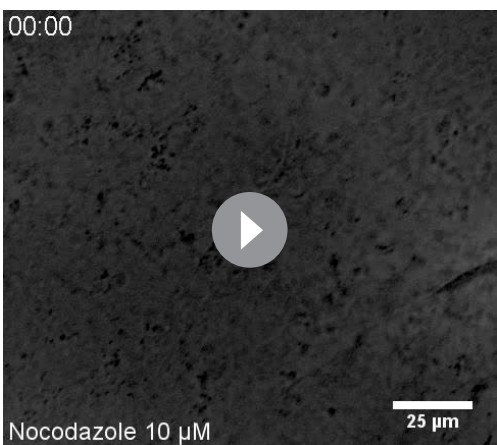

**Video 9.** Cell motility in the presence of nocodazole added at time 0, time in hh:mm, scale bar 25 µm.
https://elifesciences.org/articles/71032/figures#video9

and Golestanian, 2004; *Golestanian and Ajdari, 2008*; *Leoni and Sens, 2015*). In this case, the period of oscillation is the only timescale in the problem, and the swimming velocity must scale inversely with the period. In the limit of small oscillation amplitudes and weak hydrodynamic interactions, that is if $d \ll a \ll D \ll r$ (where $d$ is the oscillation amplitude, $a$ the bead size, $D$ the dipole size, and $r$ the cell size, *Figure 4a*), the net cell velocity over a cycle follows the scaling (*Leoni and Sens, 2015*)

$$V_{swim} = \frac{d^2}{L_s T} f_s \left( \psi \right)$$

(1)

where $f_s \left( \psi \right)$ is a periodic function of the phase shift satisfying $f_s \left( \psi = 0 \right) = 0$, that is, no velocity without phase shift, as required by the scallop theorem, and $L_s \sim r^4 / \left( a D^2 \right)$ is a length scale set by the cell geometry. Examples of theoretically computed cell trajectories in the dipole/quadrupole phase space for non-migrating and migrating cells are shown in *Figure 4—figure supplement 2*.

A key aspect of cell crawling, which is absent for micro-swimmers, is the dynamics of cell attachment and detachment from the surrounding matrix. Our observations suggest that dipole contraction is associated with an active contraction of acto-myosin clusters and that dipole extension corresponds to the elastic relaxation of the CDM following local cell detachment, that is, the loss of focal contacts (*Video 3* and *Godeau et al., 2020*). The kinetics of cell binding/unbinding to the ECM defines additional dynamic parameters, so that the scaling relationship between cell velocity and the oscillation period is less universal than *Equation 1* for the swimmer's case. An important factor is how the rate of unbinding of adhesion bonds depends on the force applied on them. At the simplest level – in the limit of fast binding kinetics and small applied force – this can be captured by a velocity scale $v_{adh} = \delta_{off} k_{off}$, where $k_{off}$ is the unbinding rate under no force and $\delta_{off}$ is a matrix deformation amplitude characterizing bond mechano-sensitivity (*Leoni and Sens, 2017*). In this case, the net velocity of the idealized cell sketched in *Figure 4a* for small oscillation amplitude reads (*Leoni and Sens, 2017*):

$$V_{crawl} = \frac{d^3}{v_{adh} L_c T^2} f_c \left( \psi \right)$$

(2)

where $f_c \left( \psi = 0 \right) = 0$ as for swimmers, and the length scale $L_c \sim r^2 / a$ also includes additional dimensionless factors related to substrate dissipations (see *Leoni and Sens, 2017*, for more details).

These simple models predict how the velocity should vary with the period of oscillation $T$. The result depends on whether the amplitude of oscillations $d$ is fixed or is a function of $T$. For oscillations of constant amplitude, the net velocity of both swimmers and crawlers decreases if the period of oscillation increases. On the contrary, the velocity is expected to increase with the period if the amplitude increases linearly with the period: $d \sim T$, as can be expected if the self-propelled object operates under constant force – or equivalently constant contraction/extension rate.

The temporal oscillations of the instantaneous cell velocity are a good readout for the dynamics of internal force generation. For such self-propelled objects, the instantaneous velocity oscillates around the average values given by *Equations 1 and 2*, with a time dependence that reflects the dynamics of the underlying force dipoles. An example of such (theoretical) velocity

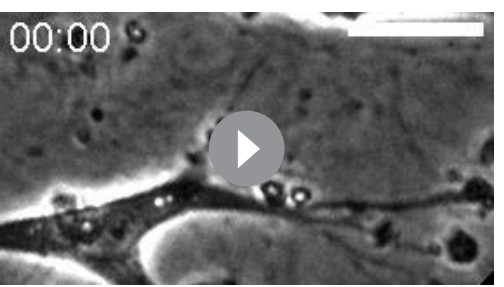

**Video 10.** Nucleus-free cell forms and shows oscillatory motion in cell derived matrix (CDM). Time in hh:mm and scale bar 25 µm.
https://elifesciences.org/articles/71032/figures#video10

oscillations can be seen in *Figure 4b*. Experimental observations indeed report strong oscillations of the instantaneous cell velocity (*Figure 4c*).

To test how the period of oscillation of migrating cells influenced their velocity, we took advantage of the inherent variability of cells and plotted the velocity vs. period for WT cells and observed a clear anticorrelation (*Figure 4d*). Although the large cellular noise prevented us from quantitatively comparing the scaling relations given by *Equations 1 and 2*, this is consistent with locomotion being driven by controlling cell deformation instead of cell traction forces. Similar conclusions have been reached in the different context of adherent epithelial cells (*Saez et al., 2005*). To extend the range of variation of the different cellular parameters, we treated cells with a collection of specific inhibitors that impact cell migration in CDMs (*Caballero et al., 2017*). We tracked cell trajectories for control cells (*Figure 4—figure supplement 1a*), cells treated with myosin ATPase activity inhibitor Blebbistatin (*Figure 4—figure supplement 1b* and *Video 11*), ROCK inhibitor Y27632 (*Figure 4—figure supplement 1c*), myosin light chain kinase inhibitor ML7 (*Figure 4—figure supplement 1d*), actin polymerization inhibitor latrunculin A (*Figure 4—figure supplement 1e*), lamellipodia growth promoter C8-BPA (*Figure 4—figure supplement 1f*), Arp2/3 complex inhibitor CK666 (*Figure 4—figure supplement 1g*), and nocodazole (*Figure 4—figure supplement 1h*). As expected, all pharmacological treatments altered cell migration. Latrunculin A (*Video 12*) and nocodazole (*Video 9*) stalled migration completely. Among the others, we observed speed oscillations for cells treated with Y-27632, ML-7, C8-BPA, and CK666. Those treatments reduced persistent cell speed (*Figure 4—figure supplement 1i*) and increased the period of speed oscillations (*Figure 4—figure supplement 1j*), showing that we could indeed alter cell migration behavior. We then plotted the velocity as a function of the oscillation period, for individual cells of drug treatments with speed oscillations aggregated (Y-27632, ML-7, C8-BPA, and CK666) together with control cells (WT). We could observe an inverse correlation between period and velocity overall for these cells (*Figure 4d*, Pearson's correlation coefficient –0.4978 with p<0.0001). However, when looking at each condition separately, persistent speed was still significantly anticorrelated with oscillation period for WT cells and Y-27632-treated cells (Pearson's correlation coefficient –0.4119 with p=0.0455 and –0.7435 with p=0.0217, respectively), but it was not in the case of ML-7, C8-BPA, and CK666. This may be caused by variability among individual cells, nevertheless on average cell speed decreased and oscillation period increased upon treatment with these compounds as well (*Figure 4—figure supplement 1i and j*). Therefore, an inverse correlation between period and velocity is evident at the population level, both for WT cells and when those are aggregated to pharmacologically treated cells.

## Externally triggered contractions by means of laser ablation induce cell translocation

Altogether, these results suggest that the temporal coupling between spatially distributed force dipoles along the cell promotes cell motion. To verify this, we designed an experiment to externally trigger localized cellular force dipoles. We observed that localized laser ablation within the cortex of a cell (*Figure 5a and b* before ablation) triggered the relaxation of the cell and the surrounding matrix, as marked by the displacement of the matrix away from the cut region (*Figure 5b* ablation). This process was then followed by a contraction of the cell along with the surrounding matrix (*Figure 5b* after ablation). Remarkably, laser ablation led to the transient recruitment of actin and myosin (*Figure 5c* bottom left and *Figure 5—figure supplement 1a*), which co-localized with cell contraction (*Figure 5c* bottom right and *Video 13*). For a polarized cell the laser-induced contraction triggered the translocation of the cell (*Figure 5b*), whereas for a cell with no clear polarization and multiple attachment points no cell body translocation was observed (*Figure 5c*). Therefore, by laser ablation we could induce matrix pinching that could lead to a cell translocation given certain cell morphology.

Mimicking the periodic alternation of contractile and extensile dipoles at both ends of the cell, a second contraction at the opposite side of the cell should lead to forward displacement. So, we then used this method to repeatedly locally impose cellular force dipoles, by triggering local contractions alternatively at the front and the back of a polarized cell (see schematics in *Figure 5—figure supplement 1b*). We imposed correlated contractions by selecting a constant time interval between consecutive laser ablations: this triggered reiterated translocation and led to a net movement of the cell (*Figure 5—figure supplement 1b* and *Video 14*). This externally induced cell displacement was achieved when laser ablation of equal power was applied sequentially at the front and the back

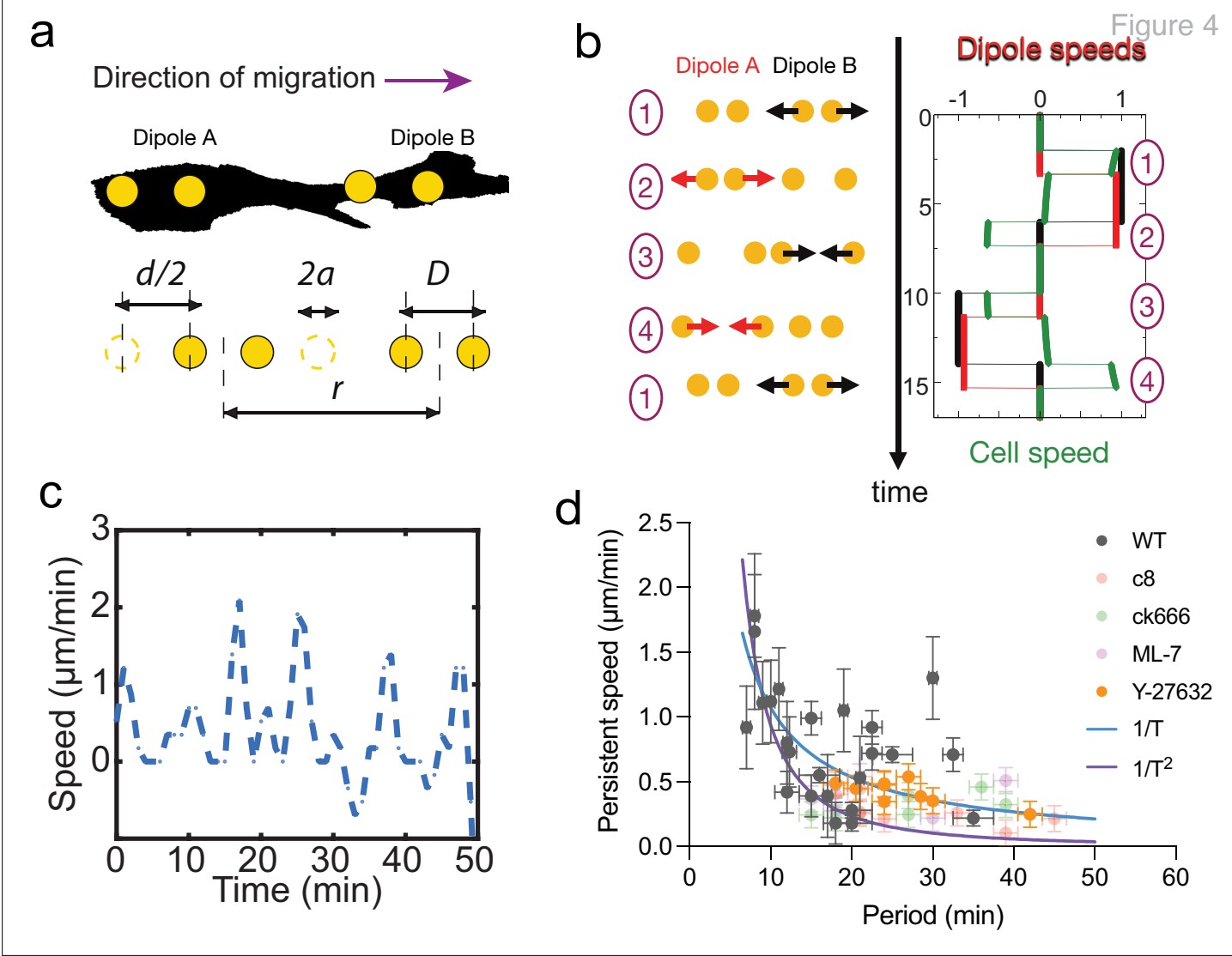

**Figure 4.** Persistent speed is related to the period of oscillations. (**a**) Schematics of dipoles distribution highlighting quantities used in the theoretical model: two dipolar units ('A' and 'B') made up of disks of radius $a$, through which cells exert traction forces on the extracellular environment. The dipoles, at distance $r$ apart, oscillate with period $T$, with minimum amplitude $D$ and a maximum amplitude $D+d$. (**b**) Model dynamics. Left: Alternate phases of extension/contraction are imposed to the two dipoles, defining a cycle ('1, 2, 3, 4, 1...') that is not time-reversible. Right: The extension/contraction rates of dipole 'A' and 'B' are shown in red and black, respectively, in unit d/T. The cell velocity, calculated using the model discussed in *Leoni and Sens, 2015*, is shown in green in the same units. It oscillates between positive and negative values – with a non-vanishing mean – with a period equal to that of individual dipoles. (**c**) Typical plot of the experimentally measured instantaneous speed of a migrating cell over time, showing oscillation with a non-vanishing mean. (**d**) Persistent speed as a function of speed period for control cells and cells treated with specific inhibitors: 10 µM ROCK inhibitor Y-27632; 10 µM MLCK inhibitor ML-7; 100 µM lamellipodia growth promoter C8-BPA, and 50 µM Arp2/3 inhibitor CK666. Error bars derived from acquisition time in x and pixel resolution in y, both divided by two. Each data point corresponds to one cell (see *Figure 4—figure supplement 1* for the number of cells). The plot displays a decay consistent with a power law. The continuous lines show the fits for $V \sim 1/T$ (dark blue) and $V \sim 1/T\,2$ (magenta), following *Equations 1 and 2* for WT cells.

The online version of this article includes the following source data and figure supplement(s) for figure 4:

**Source data 1.** Persistent speed and period of cell migration trajectories.

**Figure supplement 1.** Cell motion in cell derived matrix (CDM) is modified in the presence of specific inhibitors.

**Figure supplement 2.** Simulated cell trajectories in the dipole/quadrupole phase space.

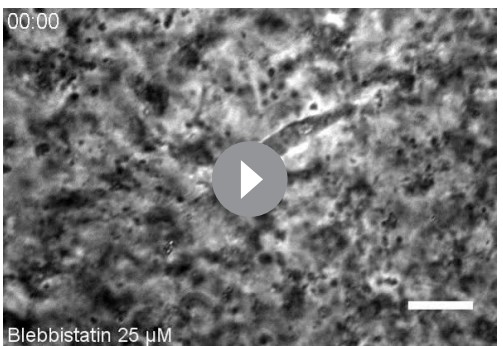

**Video 11.** Cell motility in the presence of blebbistatin added at time 0, time in hh:mm, scale bar 25 μm. https://elifesciences.org/articles/71032/figures#video11

of the nucleus, both with about 6 and 10 min intervals between cuts (*Figure 5—figure supplement 1b*). In contrast, when this sequential ablation was applied repeatedly only at one side of a polarized cell (front), net displacement was either absent or smaller than when alternating front and back, both with about 5 and 10 min intervals between cuts (*Figure 5—figure supplement 1c*). This shows that externally induced force dipoles are sufficient to promote directed cell motion and illustrates that by alternating induced force dipoles at front and back of the nucleus leads to net cell motion. Altogether, this supports the principle of a phase shift between local dipoles as a trigger for cell motility.

## Discussion

Acto-myosin complexes are likely to be the functional elements that control the dynamics of the individual contractile units. The cycles of protrusion/retraction at both ends of the cell likely involves transient retrograde actin flow whose periodic nature could be linked to a stick-slip phenomenon (*Sens, 2020*). Therefore, although the local actin dynamics within protrusions could be comparable to that fast moving cells in 2D or under confinement (*Barnhart et al., 2011*; *Liu et al., 2015*), the absence of a coherent actin flow at the scale of the entire cell and the existence of multiple competing protrusions considerably slows down cell motion. Our results show that for cells moving in 3D CDMs, the lack of time reversal symmetry required for cell motion is associated with a time shift between the oscillating dynamics of the two cell ends. Remarkably, the dynamics of individual dipoles appear similar in migrating and non-migrating cells, suggesting that the same force generation machinery is equally active in both types of cells, and that it is the synchronization between individual units that makes movement possible. Altering the dynamics of individual units can affect motion, in particular, faster oscillations can lead to faster motion, but the coordination between units is key in enabling cell translocation.

Acto-myosin networks commonly show oscillatory dynamics in vitro and in vivo in a variety of systems and over a large range of length scales (*Kruse and Riveline, 2011*): single filaments in motility assays (*Riveline et al., 1998*; *Gillo et al., 2009*; *Plaçais et al., 2009*), cells (*Negrete et al., 2016*) and cell fragments (*Paluch et al., 2005*), and even entire organisms (*Martin et al., 2009*). The biological function of these generic dynamics is often unclear. However, in our case, they appear to be essential. We conjecture that cellular systems could adapt their velocity by modulating the oscillation period. In this context, it would be interesting to test this proposal by combining tracking of single moving cells and local matrix deformations in vivo. If confirmed, this would provide an outstanding example of a physiological relevance for such oscillations. The phase shift between contractile units encodes cell polarity. Its maintenance in the course of time requires the existence of a polarity memory. If this type of phase locking can be expected in nonlinear systems (*Pikovsky et al., 2002*), it is more demanding in the cellular context, where it is challenged by strong fluctuations in protein concentrations and activities. Our results suggest that the microtubular network is involved. However, other cytoskeletal elements and their interplay with adhesion dynamics are likely to play a role as well. Disentangling the interplay between mechanics and biochemical regulation in this process remains an important open question.

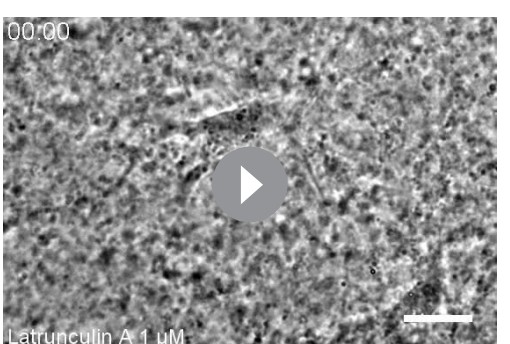

**Video 12.** Cell motility in the presence of latrunculin A added at time 0, time in hh:mm, scale bar 25 μm. https://elifesciences.org/articles/71032/figures#video12

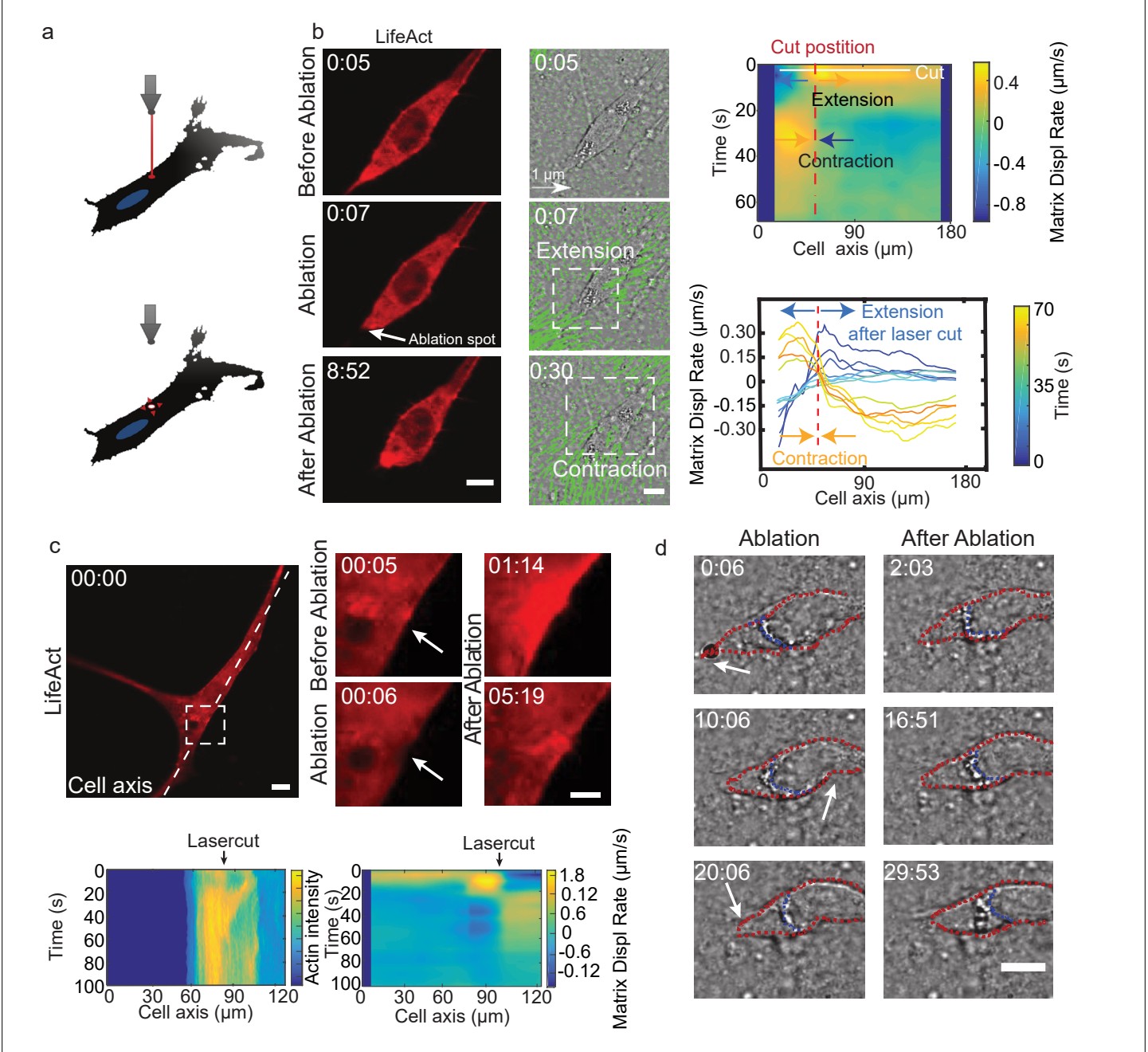

**Figure 5.** Cell motion is triggered by means of laser-induced force dipoles. (**a**) Schematic of laser ablation experiment. (**b**) (Left panel: LifeAct, middle panel: phase contrast and KLT, arrows indicate displacement between two consecutive frames, Δt=10 s). Ablation at the back of the cell (white arrow) immediately followed by an extension, and later by a contraction of the matrix (both highlighted using white square window). Scale bar LifeAct: 10 µm KLT: 20 µm. Right panel. Bottom: Plot of the displacement rate along the cell axis at different timepoints (color coded) showing extension and contraction. Top: Heatmap of displacement rates indicating the initial extension and the subsequent contraction. (**c**) Top: Sequence of snapshots during laser ablation on a cell expressing mCherry-LifeAct. Intensity drops locally immediately after the cut, followed by a local recruitment of actin, scale bar 20 µm, scale bar in zoom 5 µm. Bottom: The intensity heatmap reveals a focused actin flow (see also *Video 13*). As shown in the deformation map, the contraction precedes this flow. (**d**) Consecutive ablations (indicated with white arrows) mimic contraction-extension cycles at the front and back of the cell. Ablation is performed in the following order: at the cell back, at the front and then at the back again. In all panels, scale bar 10 µm and time in mm:ss. Note cell motion to the right (see also *Video 14*), scale bar 10 µm. The cell is outlined in red and the back of the nucleus with a blue dashed line.

The online version of this article includes the following source data and figure supplement(s) for figure 5:

**Source data 1.** Cell migration induced by laser ablation.

**Figure supplement 1.** Cell motion is triggered by means of laser-induced force dipoles.

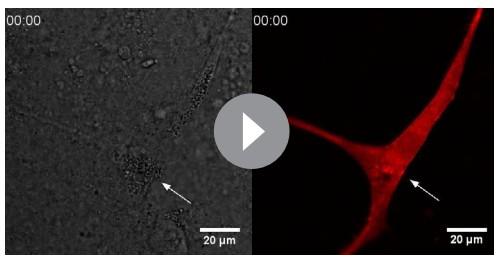

**Video 13.** Local laser ablation triggers the local recruitment of actin and local contraction, time in mm:ss.

https://elifesciences.org/articles/71032/figures#video13

Our results show that for several cell types moving in CDMs, there exist two contractile units exerting force dipoles at both ends of the cell, and the lack of time reversal symmetry required for motion is associated with a time shift between the oscillating dynamics of the two cell ends. It is interesting to note that in other reports of cell motions in 3D gels (*Steinwachs et al., 2016*), cell appears to behave as fluctuating single contractile units, with no mention of coexistence of contraction at one end and extension at the other as we report here (for instance in *Figure 1b*). This type of motion is physically possible (see for instance *Wagner and Lauga, 2013*, or *Datt et al., 2018*). It thus appears that different cell types may undergo distinct dynamics to promote directed cell motion. Single force dipoles would be related with correlation of the contractile machinery across the cell whereas two dipoles could be associated with autonomous acto-myosin contractile units on each side of the cell. It would be interesting to relate this principle of length scale of mesoscopic unit dynamics with cytoskeletal structures and compositions in different cell lines.

We have explained the mechanisms of our directed cell motion with the concept of synchronized force dipoles and our results are consistent with the associated model. However, many assumptions were made. The CDMs are not continuous media, they are hydrogels of porous nature which may halt the motion when the pore size blocks the nucleus, for example. This may contribute to phases where cells are trapped and do not move at all. This situation would not challenge our model, since the directed motion phases presumably bypass this potential blockage. Also, growth and detachments of cell adhesion at both cell ends could be involved in the migratory phases along our observations of focal contacts dynamics (*Video 3*). Again, this would not challenge the main result of our rule for finite area when cells move directionally. It may however contribute to alternative mechanisms in the model where mechanosensing at focal contacts rather than force dipoles or in synergy with force dipoles mediated by acto-myosin would contribute to the basic synchronization mechanism. These potential contributions could be disentangled by future experiments tracking both acto-myosin and focal contact proteins over time together with matrix deformations.

We propose that temporal correlations between distinct contraction-extension units along the cell body is a general principle used by mesenchymal cells to achieve directional motility in 3D. This suggests new strategies to control the motion of cells by externally modulating their local contractile activity, for which we give a proof-of-principle using a standard laser setup. This concept could also be used to design synthetic micro-crawlers. Whereas there exist many examples of artificial micro-swimmers (see *Elgeti et al., 2015*, for a review), there is to our knowledge no realization of micro-crawler in regimes where inertia is negligible.

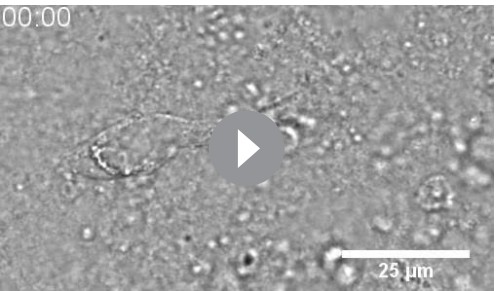

**Video 14.** Induced dipoles by local laser ablations (indicated by arrows) trigger cell motion, time in mm:ss.

https://elifesciences.org/articles/71032/figures#video14

## Materials and methods
### Cell lines
NIH3T3 MEF cell line was obtained from ATCC. REF52 and primary MEFs were obtained from the IGBMC cell culture facility. Cells lines tested negative for mycoplasma.

### Preparation of CDMs
For the CDM preparation (see *Figure 1—figure supplement 1a* and *Godeau et al., 2020*), a glass coverslip (CS) was incubated with 1% gelatin (gelatin from cold water fish skin, Sigma) and put at 37°C in the incubator for 1 hr. After two

washing steps with PBS, the CS with the gelatin solution was incubated for 20 min at room temperature with 1% glutaraldehyde (Sigma). The CS was rinsed again twice with PBS before incubation for 20–30 min with 1 M glycine (Sigma). Subsequently the CS was washed twice with PBS before plating of the NIH3T3 fibroblasts. Cells were plated at high density in order to produce CDMs. For NIH3T3, this corresponded to a cell density of $10^5$ cells/mm$^2$ in the Petri dish. The culture medium was supplemented with 50 µg/mL L-ascorbic acid and changed every 2 days. The culture was maintained for 8–9 days. Cells were removed by a lysis medium consisting of 20 mM $NH_4OH$ and 0.5% Triton (both from Sigma) in PBS after two washing steps with PBS. The pre-warmed lysis medium was carefully pipetted on the CS and incubated for up to 10 min at 37°C in the incubator. PBS solution was added and the CDM stored at 4°C. The day after, the PBS solution was carefully changed three times to remove residues of Triton. The matrices were covered with PBS and stored for up to 1 month at 4°C. For alignment purposes after image acquisition, beads (200 nm, BioSpheres) were spin-coated on the CS before incubation with gelatin. Beads for optical tweezers measurements (L4530, Sigma) were inserted when seeding the cells. For visualization of fibronectin inside the CDM, two methods were used with no apparent differences: we prepared a stable cell line expressing fluorescent fibronectin (construct kindly provided by Erickson laboratory, Duke University), alternatively, fluorescently labelled fibronectin (Cytoskeleton Inc) was added to the culture.

## Cell culture, transfection, and inhibitors

Cells were cultured at 37°C under 5% $CO_2$ with a culture medium, high glucose D-MEM with 1% penstrep (penicillin-streptomycin, Thermo Fisher Scientific) supplemented with 10% bovine calf serum (Sigma). Transfections were performed with Lipofectamin 2000 (Invitrogen) using a standard protocol, and the following constructs were used: mCherry-Lifeact, GFP-NMHC2A (nonmuscle myosin heavy chain 2A, kindly provided by Ewa Paluch lab, UCL), RFP-zyxin (kindly provided by Anna Huttenlocher lab, University of Wisconsin-Madison), or mCherry-MRLC2A (Addgene). For experiments with inhibitors, ROCK inhibitor Y-27632 was used at a concentration of 10 µM, microtubule de-polymerizing agent nocodazole at 10 µM, myosin-II inhibitor blebbistatin at 25 µM, MLCK inhibitor ML-7 at 10 µM, F-actin depolymerizing agent latrunculin A at 1 µM, Arp2/3 inhibitor CK666 at 50 µM (all from Sigma), and lamellipodia growth promoter C8-BPA at 100 µM (*Nedeva et al., 2013*). Before drug addition, we performed a control acquisition of at least 1 hr. To prevent flows, defocusing or potential damage of the CDM during manipulation, medium with drugs was added to the running experiment without removing the medium to reach the target concentration.

## Time-lapse imaging and laser ablation, optical setups, EM

We used a Nikon Ti Eclipse inverted microscope equipped with a Lambda DG-4 (Shutter Instruments Company), a charge coupled device (CCD) camera CoolSNAP HQ2 (Photometrics), a temperature control system (Life Imaging Services) and, if needed, $CO_2$ control. The objectives were the following: PhLL 20× (air, 0.95 NA, phase contrast, Nikon), a Plan Apo 60× objective (oil, 1.40 NA, DIC, Nikon) and a 40× (air, 0.95 NA, Olympus) objective with a home-made adapter to fit the Nikon microscope. Images were acquired with the NIS Elements software (v3.10, SP3, Nikon) and then exported for further processing. We also used a CKX41 inverted phase contrast microscope (Olympus) with a cooled CCD camera (Hamamatsu). The system was equipped with a temperature control (Cube box system) and 4×, 10×, 20×, and 40× phase contrast air objectives. For confocal imaging, we used a Leica TCS SP5-MP inverted microscope equipped with a Leica Application Suite Advanced Fluorescence LAS AF 2.6.3.8173/LASAF 3.1.2.8785 acquisition system with hybrid detectors (HyD), photomultiplier tube (PMT), and a heating system (Cube box system); with a HCX PL APO 63× oil objective (1.4 NA, Leica). Additionally, a Leica DMI6000 inverted microscope was used as a spinning disk, equipped with a Andor iQ 1.9.1 acquisition system, Yokogawa CSU22 spinning disk unit, and a heating system (Tokai Hit Stage Top Incubator, INU incubation system for microscopes). A HCX PLAPO 40× oil objective (1.25 NA, Leica) was used. The time between two frames ranged from 10 s to 5 min and typical exposure time was 100–200 ms. The software Imaris was used for reconstructing and animating 3D datasets. For laser ablation, a Leica TCS SP8-MP based on a Leica DM6000 CFS upright microscope was used, equipped with a Leica Application Suite Advanced Fluorescence LAS AF 2.6.3.8173/LAS AF 3.1.2.8785 acquisition system with PMT and HyD and an environmental chamber for temperature control (Cube box system). A 25× HCX IRAPO L water objective (0.95 NA, Leica) was used

and ablation performed with infrared pulsed laser (Coherent Vision II with pre-compensation). FRAP module was used with point ablation with a wavelength set at 800 nm and an exposition time of 100–200 ms. During experiments, parameters such as laser power and gain were set to minimum to have the smallest cut possible while maintaining cell and CDM integrity. 'Exploding' cells were discarded and analysis performed as described in the section referring to Image analysis.

Native CDMs with integrated cells were imaged by EM. To obtain cross-sections of the CDM, they were grown on Polystyrene sheets. The CDMs were fixed by immersion in 2.5% glutaraldehyde and 2.5% paraformaldehyde in cacodylate buffer (0.1 M, pH 7.4), and post-fixed in 1% osmium tetroxide in 0.1 M cacodylate buffer for 1 hr at 4°C. The samples were dehydrated through graded alcohol (50%, 70%, 90%, 100%), and critical point dried with hexamethyldisilazane. After mounting on stubs with conductive carbon adhesive tabs, the CDMs were coated with gold-palladium in a sputter coater (BAL-TEC SCD 005). For imaging along the z-axis, CDMs were cut and mounted upright with conductive carbon tape. Then they were examined by XL SIRION 200 FEG SEM (FEI Company, Eindhoven, The Netherlands).

## Image analysis

For quantification, the myosin intensity profiles were obtained on the raw images of the selected time frames. ImageJ was used to obtain the intensity values along a 6 μm long line, with a line width of 20 pixels to average the intensity values. Data was further normalized by the maximum value of intensity among all the frames analyzed. For visual representation, images of myosin clusters were filtered with a median filter and the background signal subtracted. To enhance the contrast of the clusters, a contrast-limited adaptive histogram equalization was applied. For KLT analysis, images were pretreated. In the case of a z-stack, images were projected with average or maximal projection and outliers were removed with Remove Outlier function of ImageJ. Then, deformation was tracked in 2D. A 'pyramidal implementation' of KLT tracker method was used to detect deformation in the mesh (*Godeau et al., 2020*). This method is based on KLT algorithm and follows bright features from one image to another. Therefore, a textured patch with high intensity variation in x and y is required. A multi-resolution pyramid of the image intensity and its gradients are computed before tracking is performed. Then, the KLT algorithm is first applied to lower resolution image, where it detects coarse movement before a higher resolution image is taken for fine movement detection. After having reached the maximum iteration steps for all pyramid levels, the displacement of the feature is extracted (between two frames). The Computer Vision Toolbox for MatLab was used with a customary written code with number of features varying between 5000 and 10,000 and a minimum distance from 8 to 14 px depending on image size and resolution, making sure that features were sufficiently spaced. Parameter window interrogation size was set to 40 px and maximal iteration to 20 px. The number of pyramids was two for all calculations. For each image an overlay of displacement vectors and phase contrast or fluorescent image of the cell was generated. Drift calculation was performed with a maximum of 40 px features with a minimum distance of 8 px with a window size of 20 px and one pyramid. The displacement due to drift was subtracted from the cell-induced displacement of the mesh.

## Analysis of the contractile and extensile patterns over time

The displacement of the meshwork calculated via the KLT feature tracker was projected onto a line going through the cell axis in order to observe 1D matrix displacement and the heatmap of displacement plotted. The matrix displacement amplitude is color coded in the heatmap. To highlight the cell position in the heatmap, cell features such as nuclear front, back, and cell tail were tracked and plotted in the heatmap. To investigate recurring contractile and extensile patterns, the divergence of matrix displacement along the cell axis was obtained. The divergence was averaged over a region of 2–5 μm wide either at the cell back or the cell front, resulting in traces for the front and the back of at least 30 min in duration with points every 30 s. Average divergence was subtracted to reduce background and autocorrelation performed via Matlab using the xcorr function with coefficient normalization so that the autocorrelation at zero lag equals 1. To determine the period, the first peak next to zero with amplitude larger than 1 standard deviation was selected. Graphs where no peak could be identified were discarded. To compare matrix displacement at two distinct positions along cell axis, cross-correlation analysis was performed with the same parameters as for the autocorrelation.

In this case the largest peak (either positive or negative) was extracted. Graphs where no peak could be identified were discarded. To obtain the period of cell speed, cell movement was tracked with the nucleus as reference point using ImageJ Manual Tracking Plugin. The trajectory was projected on the axis of migration and autocorrelation performed via Matlab. The xcorr function with coefficient normalization was used so that the autocorrelation at zero lag equals 1. The highest peak next to 0 with amplitude larger than 1 standard deviation was extracted. Graphs where no peak could be identified were discarded.

## Elasticity CDM measurements by optical tweezers

CDM with fluorescent beads were mounted on a holder and placed on an inverted microscope (Olympus IX71). A Spectra Physics YAG laser (1064 nm) was used and focused through a high numerical oil immersion objective (Zeiss achromat 100× 1.25 NA). We acquired the movies with a second objective (Olympus X40 0.6 NA associated with a CCD camera (DCC3240C, Thorlabs)). The setup was controlled by LabView 9 (National Instruments). Beads were centered in the optical trap (*Figure 1— figure supplement 1e*). The position of the CS (bottom of CDM) was registered to obtain the z position of the measured beads in the CDM. Stage was moved in 0.2 µm/s, covering a distance of 2–4 µm in x/-x and y/-y directions (*Video 2*). The laser power was calibrated with beads in solution. Subsequent data processing was performed with ImageJ bead Tracker Plugin and further post-processed with IgorPro Wavemetrics.

## Statistical analysis

No statistical methods were used to predetermine sample size. The number of experiments (*N*) and the number of cells (*n*) included in every experiment can be found at the corresponding figure caption. Individual data points are shown when possible, accompanied by the mean value and error bars corresponding to the standard error of the mean. The statistical analysis was done with GraphPad Prism, pairwise t-tests, Kruskal-Wallis tests, and one-way ANOVA for multiple comparisons were performed, and the outcomes are shown in the corresponding panels as well as indicated in the figure captions.

## Acknowledgements

We thank the Riveline Lab for discussions and help, M Maaloum, the Imaging and Microscopy Platform of IGBMC, and HP Erickson, A Huttenlocher, E Paluch for constructs, J Goetz for the former CDM protocol and M Piel for critical reading of the manuscript. DR acknowledges support from CNRS (ATIP), ciFRC Strasbourg, the University of Strasbourg, Labex IGBMC, Foundation Cino del Duca, Région Alsace, Saarland University. A Ott and D Riveline acknowledge support from DFH-UFA through the Collège Doctoral Franco-Allemand CDFA-01-13. This study with the reference ANR-10-LABX-0030-INRT has been supported by a French state fund through the Agence Nationale de la Recherche under the frame programme Investissements d'Avenir labelled ANR-10-IDEX-0002–02. ML acknowledges financial support from the ICAM Branch Contributions and Labex CelTisPhyBio No ANR-10-LBX-0038 part of the IDEX PSL No ANR-10-IDEX-0001-02 PSL. A Ott acknowledges support by DFG within the collaborative research center SFB 1027.

## Additional information

### Competing interests

Pierre Sens: Reviewing editor, *eLife*. The other authors declare that no competing interests exist.

### Funding

| Funder | Grant reference number | Author |
| --- | --- | --- |
| Deutsch-Französische Hochschule | CDFA-01-13 | Albrecht Ott Daniel Riveline |
| Deutsche Forschungsgemeinschaft | SFB 1027 | Albrecht Ott |

| Funder | Grant reference number | Author |
|---|---|---|
| Centre National de la Recherche Scientifique | | Daniel Riveline |
| ciFRC Strasbourg | | Daniel Riveline |
| University of Strasbourg | | Daniel Riveline |
| Labex IGBMC | | Daniel Riveline |
| Fondation Simone et Cino Del Duca | | Daniel Riveline |
| Region Alsace | | Daniel Riveline |
| Saarland University | | Daniel Riveline |
| Agence Nationale de la Recherche | ANR-10-IDEX-0002-02 | Daniel Riveline |
| ICAM Branch Contributions | | Marco Leoni Pierre Sens |
| Agence Nationale de la Recherche | ANR-10-LBX-0038 | Marco Leoni Pierre Sens |
| Agence Nationale de la Recherche | ANR-10-IDEX-0001-02 | Marco Leoni Pierre Sens |

The funders had no role in study design, data collection and interpretation, or the decision to submit the work for publication.

## Author contributions

Amélie Luise Godeau, Data curation, Formal analysis, Validation, Investigation, Visualization, Methodology, Writing - original draft, Writing – review and editing; Marco Leoni, Data curation, Software, Formal analysis, Validation, Visualization, Methodology, Writing - original draft, Writing – review and editing; Jordi Comelles, Data curation, Visualization, Methodology, Writing – review and editing; Tristan Guyomar, Michele Lieb, Investigation; Hélène Delanoë-Ayari, Data curation, Software, Formal analysis, Writing – review and editing; Albrecht Ott, Funding acquisition, Investigation, Writing – review and editing; Sebastien Harlepp, Data curation, Validation, Visualization, Writing – review and editing; Pierre Sens, Conceptualization, Formal analysis, Supervision, Funding acquisition, Validation, Visualization, Methodology, Writing - original draft, Writing – review and editing; Daniel Riveline, Conceptualization, Formal analysis, Supervision, Funding acquisition, Validation, Investigation, Visualization, Methodology, Writing - original draft, Writing – review and editing

## Author ORCIDs

Amélie Luise Godeau http://orcid.org/0000-0002-8778-0120
Marco Leoni http://orcid.org/0000-0002-3965-0575
Jordi Comelles http://orcid.org/0000-0002-9297-830X
Hélène Delanoë-Ayari http://orcid.org/0000-0002-8658-3942
Albrecht Ott http://orcid.org/0000-0003-0481-2658
Sebastien Harlepp http://orcid.org/0000-0001-8891-7953
Pierre Sens http://orcid.org/0000-0003-4523-3791
Daniel Riveline http://orcid.org/0000-0002-4632-011X

## Decision letter and Author response

Decision letter https://doi.org/10.7554/eLife.71032.sa1
Author response https://doi.org/10.7554/eLife.71032.sa2

# Additional files

## Supplementary files
- Transparent reporting form

## Data availability

All data generated or analysed during this study are included in the manuscript and supporting file; Source Data files have been provided for Figures 1, 2, 3, 4 and 5.

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

## Appendix 1

### Optical tweezers

The optical tweezers setup has been described in *Drobczynski, 2009*. The calibration of the stiffness as a function of the laser power was performed in PBS buffer. Two μm beads were diluted in PBS buffer and trapped. We, then, acquired a power spectrum (*Helfer et al., 2000*; *Harlepp et al., 2017*) on the quadrant photodiode and we extracted the trap stiffness. We checked the linear dependence between the stiffness and the laser power. The elastic modulus of the ECM is obtained by mixing 2 μm polystyrene beads in the matrix formation process. We follow two beads in the same field of view (*Figure 1—figure supplement 1f*). One bead is trapped in the optical tweezers whereas the second is used as a reference. Then, the piezoelectrical stage is moved and the two beads displacement is recorded on the CCD camera. A post-acquisition treatment with ImageJ allows us to extract the displacement expressed in nm of the two beads. These displacements are different one from the other due to the optical restoring force. We haven't detected any hysteresis between back and forth displacement on the same bead. This observation led us to think that the ECM in the ranges (frequency and amplitude) we were looking at was totally elastic with no viscous behavior. The knowledge of the trap stiffness allows us to determine the forces exerted on the trapped bead. The force curves obtained are linear in the regime we are exploring. We show a linear relation between the applied force and the indentation allowing us to adjust these curves with the linear model and extract the elastic modulus $E$. We performed 37 measurements at different positions in the ECM and plotted the elastic modulus obtained over the experiments. The average value of the elastic modulus that we found is around 50 Pa. We assume that the average refraction index of the ECM is close to the refraction index found in PBS. We, therefore, use the trap stiffness found in PBS to extract the forces in the ECM. We move the sample with a piezoelectrical stage at a speed rate of 200 nm/s. The displacement of the beads was recorded on the CCD camera at a framerate of 10 fps. The position of the reference bead and the trapped bead were extracted using ImageJ and the particle analysis plugin. We extracted the subpixel resolution of the center of mass positions of the beads over time and calculated the beads displacement $r = \sqrt{(x - x_0)^2 + (y - y_0)^2}$ . The displacement of the bead in the trap was linked to the force applied to the ECM, and the difference between the total displacement (reference bead) and the trapped bead was related to the ECM compression (indentation). We used several models to extract the Young's modulus from the experiments. The Hertz model described in *Nawaz et al., 2012*, and *Yousafzai et al., 2016*, were used to understand how to link the indentation to the displacement. Nevertheless, the models described in these papers were linked to single cells and not to the fibers present in the medium. We, thus, used the model described by *Laurent et al., 2002*, in the case the bead is totally immersed in an infinite 3D medium. We modified the model to introduce the indentation instead of the bead displacement in the trap that would have given a linear relation between the stiffness and the elastic modulus. Therefore, the linear relationship between the elastic modulus and the force is: $F = 2\pi REd$ with $R$ the bead radius, $E$ the elastic modulus, $d$ the indentation, and $F$ the measured force. The indentation $d$ is given by $d = d_T - d_B$, where $d_T$ is the reference bead displacement and $d_B$ the displacement of the bead in the trap.

## Appendix 2

### Multipolar expansion

As discussed in the main text, we quantify the cell-matrix interaction via the distribution of matrix displacement rate ax`round the cell, calling $u_i^{(n)}$ the component $i$ at position $n$ of the meshwork. We then calculate the monopole vector $M_i = \sum_n u_j^{(n)}$, the dipole matrix $S_{ij} = \frac{D_{ij} + D_{ji}}{2}$, where $D_{ij} = \sum_n \Delta_i^{(n)} u_j^{(n)}$ and the quadrupole matrix $Q_{ijk} = \sum_n \Delta_i^{(n)} \Delta_j^{(n)} u_k^{(n)}$, with $\Delta_i^{(n)}$ the $i$th component of the vector joining the cell center (defined below) and the point $n$ on the mesh. We also compute the characteristic scale of matrix displacement rate defined as $\sum_n \left| u^{(n)} \right|$. The largest component of the monopole vector is called the main monopole, $M$, and the largest eigenvalues of the dipole and quadrupole tensors are defined as the main dipole, $D$, and the main quadrupole, $Q$. The corresponding eigenvectors are defined as the main dipole and quadrupole axes.

A cell that migrates spontaneously (i.e. in absence of external forces) can be viewed as a force-free body, hence in the absence of inertia the sum of all the traction forces add up to zero (**Tanimoto and Sano, 2014**). Thus, for a homogeneous material under linear elasticity, the monopole of displacement rate should be zero. In our experiments, 3D imaging is challenged by a limited and asymmetric field of view which leads to a non-zero value of the monopole. To reduce these imaging asymmetries, the traction, monopole, dipole, and quadrupole are computed and averaged over disks (**Figure 3—figure supplement 1**) of increasing radius $R_k$, $k = 0, 1, ...M$ starting from a minimum radius $R_0 \simeq 20 \mu m$ and up to a maximum radius $R_M$ defined as the largest radius such that the circular region is fully contained within the boundaries of the experimental images. To minimize the spurious monopole, the center of the disk is varied around the apparent cell center (obtained from cell tracking) and the location for which the monopole is minimum is adopted as the center of the disks on which the dipoles and quadrupoles are computed. This is done at every time step. Note that matrix heterogeneities could also possibly contribute to a non-zero monopole. **Figure 3—figure supplement 2** shows histograms for difference of orientation between the main dipole axis and the direction of motion for three examples of migrating cells, showing a clear peak near zero angle difference. In addition to cellular noise, the rather large spread of these data is also explained by the fact that the major dipole is a fluctuating quantity which frequently crosses zero, at which point the largest dipole is the minor dipole, which is typically oriented perpendicular to the direction of motion.

Additional examples of cycles for migrating and non-migrating cells are shown in **Figure 3—figure supplement 3a**. As in the main text, the cycles shown for migrating cells are obtained by first identifying intervals of time with clear oscillating behavior in the speed and focusing on that time interval when we compute the multipolar expansion. For cells which are non-migrating, we cannot apply this scheme as we have no notion of oscillating speed, thus the choice in **Figure 3—figure supplement 3a** is somehow arbitrary. Nonetheless, we find that the areas of cycles for non-migrating cells are systematically smaller when compared to the areas of migrating cells. Importantly, the comparison between the absolute values of the multipoles of displacement rates between different cells is not straightforward, as these measures depend on the spatial and temporal resolution of the movies, which may vary from cell to cell. To support our claim that the main difference between migrating and non-migrating cells is the existence of a phase shift and not the cell's ability to exert traction forces, we show in **Figure 3—figure supplement 3b** the average of the absolute value of the rate of deformation field for migrating and non-migrating cells. This quantity is obtained for each cell by computing the sum of the displacement rate amplitude within a disk, divided by the number of measurement points within the disk, averaged over disk radii spanning the accessible range for each cell, and averaged over time. Although this quantity varies from cell to cell, there is no significant difference (p=0.2766) between migrating and non-migrating cells. On the other hand, the area enclosed by the cycle in the dipole/quadrupole phase space is systematically higher for migrating than non-migrating cells. **Figure 3—figure supplement 3c** shows the cycle area, defined as the area $A(n_1, n_s) = \sum_{i=n_1}^{n_2-1} I_{i+1,i} + I_{n_1,n_2}$ with $I_{i+1,i} = \int_i^{i+1} Q(D) \, dD = \frac{1}{2} (Q_{i+1} + Q_i)(D_{i+1} - D_i)$, where $D_i$ and $Q_i$ are the dipole and quadrupole values at the $i$th time frame and $n_1$ and $n_2$ are the starting and ending time frames. Both the absolute value of this quantity in physical units and the fraction (in %) of normalized the area of the rectangle tightly enclosing the cycle are shown.

