## [Editor Report]

This manuscript suggests a novel mechanism by which an animal cell can move through a three-dimensional extracellular matrix, namely by synchronized oscillations in contraction at the front and at the back of the cell. It should be of great interest to a variety of researchers in cellular biophysics.

---

## [Decision Letter]

**Decision letter after peer review:**

Thank you for sending your article entitled "Temporal correlation between oscillating force dipoles drives 3D single cell migration" for peer review at *eLife*. Your article is being evaluated by 2 peer reviewers, and the evaluation is being overseen by a Reviewing Editor and Aleksandra Walczak as the Senior Editor. We apologize for the delay in furnishing this report.

Given the list of essential revisions, including new experiments, the editors and reviewers invite you to respond as soon as you can with an action plan for the completion of the additional work. We expect a revision plan that under normal circumstances can be accomplished within two months, although we understand that in reality revisions will take longer at the moment. We plan to share your responses with the reviewers and then advise further with a formal decision.

While both reviewers are very supportive of the work, it was clearly felt that the experimental data was not sufficiently convincing. This suggests additional experiments may be warranted, or at the very least more analysis of the data is called for, especially in light of the fact that the underlying theory has already been published elsewhere.*Reviewer #2 (Recommendations for the authors):*

The manuscript proposes a model of cell motion in which the cell front and back are performing the same kind of contraction-extension cycles. When these two cycles are in antiphase, this results in back-and-forth motion while a phase difference between the two cycles not equal to zero or π results in cell migration i.e. a non-zero average velocity. This is an interesting model which differs from the usual views of cell motion coming from the treadmilling of actin in a lamellipod or by extension of pseudopods at the cell front. However, I am not fully convinced that the presented evidence supports this model rather than alternative ones.

The main results of the manuscript are obtained by observing and comparing the matrix deformations for migrating fibroblasts and for non-migrating ones that undergo back and forth motion. These observations as well as theoretical arguments led the authors to conclude that cell migration is associated to periodic localized contractions at the front and back of the cell with a finite phase difference. I find it difficult to assert from these data only that the contraction processes are similar at the front and back of the cells. For instance, the data seem similar in many respects, to those of Tanimoto and Sano (2014) obtained for 2D amoeboid motion on a substrate.

I list below more specific concerns with the presented data.

The assay which uses extracellular matrix deposited by confluent cells seems adequate and the quantification of matrix deformation soundly performed. I am less convinced by the presented quantification of the data in Figure 2 and 3.

Figure 2. The autocorrelations of the deformations in Figure 2d and 2e clearly show their periodic character. I guess these are representative curves for a single cell but this is not clearly stated. No details are given on how it was decided that an autocorrelation was periodic and how the period was extracted. The duration of the time traces that were used to compute these auto correlations is not stated either. The number of cells/observations used for the quantitations of the periods are not provided. The crucial panels j and k are even less clear for me. I clearly see from Figure 1j that matrix deformations at the front and back are in antiphase for the non-migrating cells. I also see that the cross-correlation is non symmetric in time for the migrating cell but I am unable to see how a phase lag was extracted for the presented example (there clearly seem to be a negative peak at zero). All this would need to be precisely explained to assess these results.

Figure 3. In (c) for the non-migrating cell the presented time trace for the dipolar and quadrupolar deformation are clearly in antiphase which nicely agree with Figure 1j.

However, for the migrating cell I again do not see a definite phase lag for the two traces. It rather seems for this particular trace, that the dipolar and quadrupolar deformations do not oscillate at the same frequency.

Figure 3d-h illustrate the cycle with a finite area in the dipole-quadrupole plane but appear as anecdotal examples. However, this should be quantified (average area, perhaps correlated with some characteristic of the motion since I guess cell may change direction).

Figure 4. I am not sure to understand the logic behind this experiment nor exactly what to conclude from it. First, when one considers separately each category of cells (untreated, treated with a given compound) it seems that there is no correlation between the period and the speed (is this the case?). This indicates that the variability between individual cells is large. Then, it would seem better to compare the effect of each treatment at the level of individual cells i.e. for given cell how the treatment change its period of contractions and its speed. Is this compatible with (VT^n)_(before treatment) = (VT^n)_(after treatment). If this equality is not obtained, this would indicate that other parameters are affected by the treatment and the need to determine them before concluding anything (in the framework of the authors, it would for instance be useful to assess whether the dephasing psi is modified by the treatment).

Figure 5. Artificially-induced contractions produced by laser ablation are taken as further evidence in favor of the model. This is obviously very different from physiological motion but in addition, I am not sure how this really fits with the proposed model. As far as I can see, in the example provided in Figure 5h, the ablation at the back and front were in antiphase, which should produce back and forth motion rather than directed migration. Again here, one would like convincing quantification (distance travelled given number of cycles of front/back ablations) and the dependence on important parameters like the dephasing between cycles of ablation at the cell back and front.

I do not have specific suggestions beyond those made in the public review but I like to add one additional point. The dipolar deformation is a 3X3 matrix. The authors computed its 2D projection but made use only of its projection on the direction of average cell velocity. I would welcome a discussion of the respective directions of the cell velocity and the main axis of the projected 2D dipolar deformation matrix.*Reviewer #3 (Recommendations for the authors):*

The authors have studied fibroblasts migrating in cell-derived matrix (CDM), which is a fibrous three-dimensional ECM with fluorescently labeled fibronectin and large pores. Cell were divided into two classes, non-migrating and migrating ones. In both cases, phases of contraction and extension were observed at the front and at the back (separated by the nucleus), with a typical oscillation period of 10 min, but the migrating cells showed a time lag, while the non-migrating ones did not. The authors argued that this phase shift (or phase locking) is the hallmark of migration, because it breaks the time-reversal symmetry in the phase space defined by deformation dipole and quadrupole (because the matrix is shown to be elastic by laser optical tweezer experiments, this should correspond to a cycle in force dipole and quadrupole space). In analogy to the scallop theory for microswimmers (equation 1, from Leoni and Sens 2015), the authors argue that only a non-vanishing phase shift leads to a finite crawling velocity (equation 2, from Leoni and Sens 2017). The have performed two experiments to validate this concept. With a large range of pharmacological inhibitions, they show that crawling speed is inversely proportional to oscillation period, as predicted by the theory. By using laser ablation to generate force contraction dipoles, they show that cell motion can be induced by alternating dipoles.

The big strength of this work is the elegance of the theoretical concept and the creativity in analyzing a well-known system from a new and quantitative angle. The weakness of this work is that the experimental data is not entirely convincing and that open questions remain regarding the generality of this concept.

I first note that three-dimensional traction force microscopy in collagen gels (as done e.g. by the group of Ben Fabry, often for cancer cells similar to the ones studied here) usually shows a clear dipole. Here the authors suggest that if one looks closely, there are actually two dipoles, separated by the nuclear region. I wonder if this observation depends on the matrix used (CDM with large holes) and the cell type used.

Next the authors describe that the two dipoles show oscillations which are phase-shifted in the migratory case. It is not entirely clear what causes these oscillations. Figure 1c and d suggest a relation to myosin and microtubules, but here the image quality is very bad and no quantification is being given. Figure 4d seems to suggest that oscillations and migration persist even under severe perturbations. Therefore it remains unclear what causes these oscillations and if they are really needed for productive cell migration.

The phase plane plot with dipoles and quadrupoles in Figure 3 is very interesting and similar to earlier work on *Dictyostelium* migrating on a two-dimensional substrate (Tanimoto and Sano BPJ 2014). I wonder why the same plot is not being shown for the theory part of Figure 4, because if it agreed, it would strengthen the relation between theory and experiment. Moreover the work by Tanimoto and Sano raises the question why the authors do not apply their concept to two dimensions. They argue in the discussion that in two dimensions, actin flow breaks the front-back symmetry and thus another mechanism is needed in three dimensions. This brings up two questions. First is the approach by Tanimoto and Sano for two dimensions not valid and second what is the proof that there is no retrograde flow in three dimensions.

Finally, it is really hard to assess if the laser ablation experiments do support the concept that oscillating dipoles leads to cell migration. Obviously, they not only cause relaxation and reformation of contractile structures, but they also interfere with adhesion and one effect might be to simply cause recoil in an elastic matrix.

In summary, the theoretical concept is very appealing and the experiments conducted are creative and interesting, but the relation between theory and experiment is not entirely convincing. Therefore, impact in the wider community of quantitative life sciences might remain limited for the current form of this manuscript.

In order to improve the relation between the general concept and the supporting experimental data, it would be very informative to conduct these experiment with different matrices and different cells types (ideally even with ghosts without nucleus), to see how general this observation is. How much does pore size of CDM matters, both for the oscillations and the laser ablation experiments? Is it important that the nucleus separates these two regions of the cell? Why cannot the same experiment be one in two dimensions, where the nucleus is out of the way? It would also be interesting to consider if less perturbative experiments might be feasible (e.g. optogenetics to locally induce contractility).

For the theory part, it would be interesting to give more background on the calculations behind Equation 2 and to explain if it also generates an open loop in the D-Q-plane as shown here for the experimental data.

Finally, I note that the discussions of the scallop theorem for swimming, although interesting, are not directly relevant to this work, because as the authors comment themselves, the situation is very different for crawling. Of course, this is an interesting analogy, but it also might be distracting or confusing to the general readers of *eLife*; it would be more interesting in the context of a physics journal. It is somehow odd that there are more references on the theory of swimming than on the theory of crawling. I note that there is actually a literature on the scallop theorem for crawling cells not cited here, most prominently the work by Pierre Recho, compare Recho, P., J-F. Joanny, and L. Truskinovsky. "Optimality of contraction-driven crawling." Physical Review Letters 112.21 (2014): 218101. I also miss references to earlier use of force multipoles in the cellular context (here mainly the work of Tanimoto and Sano is cited), and to traction force microscopy (including the work by Ben Fabry), which can be used to measure them.

[Editors' note: further revisions were suggested prior to acceptance, as described below.]

Thank you for resubmitting your work entitled "Temporal correlation between oscillating force dipoles drives 3D single cell migration" for further consideration by *eLife*. Your revised article has been evaluated by Aleksandra Walczak (Senior Editor) and a Reviewing Editor.

The manuscript has been improved but there are some remaining issues that need to be addressed, as outlined below:

1) The revised submission is strongly improved over the initial submission. The authors have conducted new experiments (other matrix, other cell types, laser ablation), added more theory results (in particular the D-Q plots), improved image quality and analysis, and made the main text more accessible. In my view, however, there still remains some doubt about the general validity of these results. As the authors write themselves, cell migration in 3D matrices can also occur through coherent actin flows or with single rather than double dipoles, so the title "Temporal correlation between oscillating force dipoles drives 3D single cell migration" and the corresponding abstract still sound too general. Interestingly, there now seem to be at least two scallop theorems for crawling, one for actin flow by Recho and coworkers and one for the contractile oscillations described here. We suggest that it is erroneous to now only citie Recho PRL 2013, because it is really the PRL 2014 that discusses the scallop theorem.

2) The manuscript has improved with more quantification available to assess the results and more precise explanations of how the results were obtained. The quantified data appear to support the interesting finding that there are two extension/contraction dipoles at the front and back of the cell. The dipole/quadupole cycle differences (Figure 3) also appear convincing, in agreement with the previous work of Tanimoto and Sano.

Yet, there remain several further points that are not fully convincing. The first main one is the dephasing between the two dipoles which is based on a single peak in the cross correlation and appears very noisy (the statistical significance of the count statistics Figure 2k should at least be provided).

Second, we are uneasy with the conclusions drawn for the laser ablations (Figure 5) and still do not see how they support (or not) the model. For instance, repeated ablations in the front of the cell would not lead to cell translocation given the model advocated by the authors (if we understand correctly). But this control has not been performed. Moreover, given the authors' model one would expect some dependence on the relative phase of ablations in the front and the back. But this is not attempted. We are thus unsure if one can conclude something very definite from these ablation experiments.

3) Overall, the reviewers are of the option that the study is strongly theory-driven and that the authors could do better in describing the experimental evidence. The whole design of the manuscript is such that the idea is put first and then some experimental evidence is provided. An alternative approach would be to start with the general status of the field of cell migration, then introduce the experimental results for this specific and certainly interesting system, and then the theory explanation as one way to explain these observations. Please consider this possible revision.

---

## [Author Response]

Reviewer #2 (Recommendations for the authors):The manuscript proposes a model of cell motion in which the cell front and back are performing the same kind of contraction-extension cycles. When these two cycles are in antiphase, this results in back-and-forth motion while a phase difference between the two cycles not equal to zero or π results in cell migration i.e. a non-zero average velocity. This is an interesting model which differs from the usual views of cell motion coming from the treadmilling of actin in a lamellipod or by extension of pseudopods at the cell front. However, I am not fully convinced that the presented evidence supports this model rather than alternative ones.

We thank the reviewer for acknowledging the interest and originality of our model and hope that the improvement to the manuscript detailed below will convince the referee.

The main results of the manuscript are obtained by observing and comparing the matrix deformations for migrating fibroblasts and for non-migrating ones that undergo back and forth motion. These observations as well as theoretical arguments led the authors to conclude that cell migration is associated to periodic localized contractions at the front and back of the cell with a finite phase difference. I find it difficult to assert from these data only that the contraction processes are similar at the front and back of the cells. For instance, the data seem similar in many respects, to those of Tanimoto and Sano (2014) obtained for 2D amoeboid motion on a substrate.

We thank the reviewer for raising this point. We provide more quantification of the oscillations at the back and front of the cell, which happen to show no significant differences between the two sides (Figure 2 Supplementary Figure 1 and Figure 2 Suppl. Figure 2 and Results section p. 4). It is worth noting that from the theoretical perspective, front and back ‘contractions’ do not need to be similar to elicit net cell translocation. Indeed, we did not include analysis of the magnitude of the contractions since phase difference is sufficient to break symmetry. This is now discussed explicitly in the text (p. 6).

Regarding the novelty compared to Tanimoto and Sano 201 : their paper on *Dictyostelium discoideum* crawling on deformable 2D substrate (Tanimoto and Sano, 2014) was important regarding the use of multipole analysis to identify the existence of a finite cycle in a well chosen phase space. Here, in addition to the 3D nature of motion in a physiological environment (the CDM), we substantiate the lack of time reversal symmetry as originating from: (i) the existence of (at least) two contractile units in the cell, and (ii) the appearance of a phase shift between their dynamics. Furthermore, we make the connection between the force dipoles and cellular readouts, *i.e.* myosin cluster formation and contraction.

I list below more specific concerns with the presented data.The assay which uses extracellular matrix deposited by confluent cells seems adequate and the quantification of matrix deformation soundly performed. I am less convinced by the presented quantification of the data in Figure 2 and 3.Figure 2. The autocorrelations of the deformations in Figure 2d and 2e clearly show their periodic character. I guess these are representative curves for a single cell but this is not clearly stated. No details are given on how it was decided that an autocorrelation was periodic and how the period was extracted. The duration of the time traces that were used to compute these auto correlations is not stated either. The number of cells/observations used for the quantitations of the periods are not provided.

The reviewer is correct that these curves are for single cells and are statistically representative. This is now stated in the revised version (p. 4) when the panels are quoted in the text. We agree that the process followed to determine periodicity in autocorrelation functions needed clarification. We now detail how the correlation functions were obtained and analyzed including the specific points raised by the reviewer. The process is now fully explained in the methods section “Analysis of the contractile and extensile patterns over time”, p. 13. Moreover, we added a new supplementary figure (Figure 2 – supplementary figure 1) that shows the analysis pipeline. The duration of the traces was at least 30 minutes with time points every 30 seconds. When traces are shown, their duration is stated in the Figure caption. The number of cells used is also provided in captions. The criteria for determining the period was the existence of a peak larger than 1 standard deviation. Finally, we added a section for the statistics (p. 14).

The crucial panels j and k are even less clear for me. I clearly see from Figure 1j that matrix deformations at the front and back are in antiphase for the non-migrating cells. I also see that the cross-correlation is non symmetric in time for the migrating cell but I am unable to see how a phase lag was extracted for the presented example (there clearly seem to be a negative peak at zero). All this would need to be precisely explained to assess these results.

The Phase lag in panel j, which is an example, is obtained from the position of the peak of largest amplitude. We agree with the referee that these panels are of key importance, and we now clarify this analysis better. To do so, we clarified the panels 2f, j and h indicating the peaks considered in these clearer examples. Moreover, the criteria followed have been further clarified in the methods section (Analysis of the contractile and extensile patterns over time). Also, another example of individual correlation functions that leads to the distribution of Figure 2k is shown in the new Figure 2 —figure supplement 1, that acts as guide for the analysis pipeline as well.

Figure 3. In (c) for the non-migrating cell the presented time trace for the dipolar and quadrupolar deformation are clearly in antiphase which nicely agree with Figure 1j.However, for the migrating cell I again do not see a definite phase lag for the two traces. It rather seems for this particular trace, that the dipolar and quadrupolar deformations do not oscillate at the same frequency.

We thank the reviewer for raising this point. The cellular system is noisy, and it is indeed not straightforward to extract an average phase lag for migrating cells in (c). To address this, we used the phase space representation of panel (d), (e), (h), and in the other examples shown in Figure 3 —figure supplement 3. This representation is more appropriate to reveal the mean phase lag through the existence of a cycle with finite area unlike what is seen for non-migrating cells. As the referee pointed out, this is an important element that is now discussed in the new version (p. 5).

Figure 3d-h illustrate the cycle with a finite area in the dipole-quadrupole plane but appear as anecdotal examples. However, this should be quantified (average area, perhaps correlated with some characteristic of the motion since I guess cell may change direction).

As the referee indicates, we provided several examples of the dipole-quadrupole analysis. We agree that further quantification was needed. In the revised version of the manuscript, we show the quantification of areas enclosed by the trajectories of migrating and non-migrating cells in the D-Q plane. The results are shown in Figure 3 —figure supplement 3 (with technical explanations in Appendix 2) and discussed in the main text (p.6). The area is systematically larger for migrating cells. In order to compensate for possible variation of dipole and quadrupole intensity between migrating and non-migrating cells, we also show a normalized area independent of the absolute values of the multipoles (explained in Appendix 2), which also show a systematic difference between the two cellular behaviors.

Figure 4. I am not sure to understand the logic behind this experiment nor exactly what to conclude from it. First, when one considers separately each category of cells (untreated, treated with a given compound) it seems that there is no correlation between the periodand the speed (is this the case?). This indicates that the variability between individual cells is large. Then, it would seem better to compare the effect of each treatment at the level of individual cells i.e. for given cell how the treatment change its period of contractions and its speed. Is this compatible with (VT^n)_(before treatment) = (VT^n)_(after treatment). If this equality is not obtained, this would indicate that other parameters are affected by the treatment and the need to determine them before concluding anything (in the framework of the authors, it would for instance be useful to assess whether the dephasing psi is modified by the treatment).

As the referee points out, the drug experiments that we performed did not intend to identify a molecular actor involved in the migration process. On the contrary, we used the different drug treatments as a tool to perturb cell speed and relate it to changes in period as the model suggests. We have studied the correlations between persistence speed and period for wild type and for different drug treatments, both as a whole and separately. An inverse correlation between period and speed can be statistically demonstrated when cells for all conditions are aggregated (Pearson’s coeff -0.4978 with significance p < 0.0001). When this anticorrelation was analyzed individually for each condition, we found that it still held for WT cells (Pearson’s coeff -0.4119 with significance p = 0.0455). Among the different drugs we tried, only Y27632 showed a significant anticorrelation at the population level (Pearson’s coefficient -0,7435 with p = 0.0217). The other drugs tested (C8, CK666 and ML-7) did show correlations but were not statistically significant (C8) or showed no correlation (CK666 and ML7). We therefore fitted the expected power-law relationship between period and velocity only using WT treatment (modified Figure 4d). Nonetheless, on average all treatments showed a reduction of speed and an increment of oscillation period, suggesting that there is indeed a high variability among individual cells as indicated by the referee. We therefore tested the single cell analysis suggested by the reviewer before and after treatment. Due to the low number of cells that were migrating both before and after the treatment, the results were affected by the inherent variability of the experimental system and results were not conclusive.

However, this does not challenge our main conclusions that the 3D motility of our cells is based on phase shifted oscillating force dipoles. This claim is substantiated by the systematic existence of a phase shift and a cycle of finite area for motile cells and its systematic absence for non-motile cells (Figure 2k and the new quantification of cycle area in Figure 3 Suppl. Figure 3 b). The results of Figure 4 attempt to deepen our understanding of the motility strategy by analyzing correlations between speed and oscillation period and comparing our results with theoretical models. A clear anticorrelation exists for WT cell which allows us to conclude that locomotion being driven by controlling cell deformation instead of cell traction forces.

We thank the reviewer for pointing out that the use of drugs does not merely allow to travel in the parameter space proposed by the theoretical model but can have complex effects which are clearly beyond the scope of the present paper. These points are now discussed in the new version of the text p. 8.

Figure 5. Artificially-induced contractions produced by laser ablation are taken as further evidence in favor of the model. This is obviously very different from physiological motion but in addition, I am not sure how this really fits with the proposed model. As far as I can see, in the example provided in Figure 5h, the ablation at the back and front were in antiphase, which should produce back and forth motion rather than directed migration. Again here, one would like convincing quantification (distance travelled given number of cycles of front/back ablations) and the dependence on important parameters like the dephasing between cycles of ablation at the cell back and front.

We agree that further quantification of the effect that the artificially-induced contractions have in migration will help in strengthening the model. We added new laser ablation experiments (see new Figure 5 —figure supplement 1). In particular, we performed multiple ablations: the second ablation occurs while the first ablation and myosin enrichment has not yet relaxed, and we expected the two induced contractions to interact and help motion. This was specifically observed and this result is now reported. We quantified and analyzed cell migration with these artificially-induced contractions (see new Figure 5 —figure supplement 1) and we now discuss this within the framework of the model p. 9.

I do not have specific suggestions beyond those made in the public review but I like to add one additional point. The dipolar deformation is a 3X3 matrix. The authors computed its 2D projection but made use only of its projection on the direction of average cell velocity. I would welcome a discussion of the respective directions of the cell velocity and the main axis of the projected 2D dipolar deformation matrix.

These two axes tend to be aligned although not exactly because of noise. We now discuss in p. 5 this point in the new version with added data on the distribution of the angle between the two axes (Figure 3 —figure supplement 2).

Reviewer #3 (Recommendations for the authors):The authors have studied fibroblasts migrating in cell-derived matrix (CDM), which is a fibrous three-dimensional ECM with fluorescently labeled fibronectin and large pores. Cell were divided into two classes, non-migrating and migrating ones. In both cases, phases of contraction and extension were observed at the front and at the back (separated by the nucleus), with a typical oscillation period of 10 min, but the migrating cells showed a time lag, while the non-migrating ones did not. The authors argued that this phase shift (or phase locking) is the hallmark of migration, because it breaks the time-reversal symmetry in the phase space defined by deformation dipole and quadrupole (because the matrix is shown to be elastic by laser optical tweezer experiments, this should correspond to a cycle in force dipole and quadrupole space). In analogy to the scallop theory for microswimmers (equation 1, from Leoni and Sens 2015), the authors argue that only a non-vanishing phase shift leads to a finite crawling velocity (equation 2, from Leoni and Sens 2017). The have performed two experiments to validate this concept. With a large range of pharmacological inhibitions, they show that crawling speed is inversely proportional to oscillation period, as predicted by the theory. By using laser ablation to generate force contraction dipoles, they show that cell motion can be induced by alternating dipoles.The big strength of this work is the elegance of the theoretical concept and the creativity in analyzing a well-known system from a new and quantitative angle. The weakness of this work is that the experimental data is not entirely convincing and that open questions remain regarding the generality of this concept.I first note that three-dimensional traction force microscopy in collagen gels (as done e.g. by the group of Ben Fabry, often for cancer cells similar to the ones studied here) usually shows a clear dipole. Here the authors suggest that if one looks closely, there are actually two dipoles, separated by the nuclear region. I wonder if this observation depends on the matrix used (CDM with large holes) and the cell type used.

We thank the reviewer for acknowledging the originality of our work. We agree that further data and analysis do now strengthen our manuscript. As requested by the reviewer #2, we show in more detail the existence of the two dipoles with further quantification of their magnitude (Figure 2 —figure supplement 2). As the referee points out, the existence of two dipoles differs from previous work on collagen gels. We now discuss this further in the revised version of the manuscript. Moreover, we show how other cell types and cytoplasts (cell fragments lacking nucleus) migrate in our set-up (Figure 1 —figure supplement 3 and Video 10).

We also elaborate further in the discussion about the two dipoles we report and its potential difference with the single dipole per cell reported by Ben Fabry in articles. For example, in Steinwachs et al., Nature Methods 13:171 (2016) https://doi.org/10.1038/nmeth.3685, single dipoles are visible throughout the paper and for different matrices (collagen, fibrin, Matrigel) and for different collagen concentrations. Pore sizes are similar to our work. So, the cell type could be involved in the phenomena. It is worth noting that we also now show measurements of these two dipoles for two more cell types (primary mouse embryonic fibroblasts and REF52 cell line). We agree with the reviewer that the two dipole features could depend on the cell lines and their inherent ability to contract autonomously on each side of the nucleus. This point is now reported in the revised version p.11.

Next the authors describe that the two dipoles show oscillations which are phase-shifted in the migratory case. It is not entirely clear what causes these oscillations. Figure 1c and d suggest a relation to myosin and microtubules, but here the image quality is very bad and no quantification is being given.

We agree with the referee. We increased the quality of images in Figure 1c-d accordingly. We treated the images in order to facilitate the visualization. Microtubule cytoskeleton is shown now in a clearer manner. Clarity of myosin clusters during the contraction process has been improved as well. Moreover, we added a new vidoe (Video 6) to show how the myosin signal densifies over time and vanishes afterwards. This new vidoe goes along with new quantification of the process, which is given in new Figure 1c.

Figure 4d seems to suggest that oscillations and migration persist even under severe perturbations. Therefore, it remains unclear what causes these oscillations and if they are really needed for productive cell migration.

As we wrote for reviewer 2, the drug experiments that we performed did not intend to identify a molecular actor involved in the migration process. We used the different drug treatments as a tool to perturb cell speed and relate it to changes in period as the model suggests. We now clarify this point further in the Results section. Moreover, correlation coefficients between velocity and oscillation period along with their statistical significance are now reported as well (on p. 8). Please refer to our answer to reviewer 2 on this point for a detailed explanation. Briefly, the sustained motion in the presence of the drug does not challenge the main mechanism: it suggests that cells adopt alternative strategies to keep the generic oscillatory patterns and perform persistent motion.

The phase plane plot with dipoles and quadrupoles in Figure 3 is very interesting and similar to earlier work on *Dictyostelium* migrating on a two-dimensional substrate (Tanimoto and Sano BPJ 2014). I wonder why the same plot is not being shown for the theory part of Figure 4, because if it agreed, it would strengthen the relation between theory and experiment.

We thank the referee for the suggestion. We now add the phase plane plot for the theory (Figure 4 —figure supplement 2), which indeed strengthens the relation between theory and experiments by showing a cycle with finite area for migrating cells and a vanishing area for non-migrating cells.

Moreover, the work by Tanimoto and Sano raises the question why the authors do not apply their concept to two dimensions. They argue in the discussion that in two dimensions, actin flow breaks the front-back symmetry and thus another mechanism is needed in three dimensions. This brings up two questions. First is the approach by Tanimoto and Sano for two dimensions not valid and second what is the proof that there is no retrograde flow in three dimensions.

The point raised by the referee is very important. We are not saying that the motility strategies in 2D and 3D are different, one consisting of retrograde flow and the other not. We propose that these two strategies can exist in any dimension. Therefore, we certainly believe that Tanimoto and Sano approach is valid.

We do not claim there is no retrograde flow at all for cells moving in our CDM: cells form protrusions and this process should involve some local retrograde flow. We claim that there is no retrograde flow from the front to the back of the cell as reported in former studies quoted in the article but instead growing and shrinking protrusions can be seen at the front and at the back of cells. This is further discussed in the revision (p. 9).

Finally, it is really hard to assess if the laser ablation experiments do support the concept that oscillating dipoles leads to cell migration. Obviously, they not only cause relaxation and reformation of contractile structures, but they also interfere with adhesion and one effect might be to simply cause recoil in an elastic matrix.

We added more experiments of laser ablation along with the quantification of the induced migration (see new Figure 5 —figure supplement 1 and Results section p. 9). The matrix recoil does occur immediately following laser ablation and lasts a few seconds, but it is followed by myosin recruitment and active contraction at a longer time scale of about a minute (Figure 5b). This substantiates that local dipoles are induced within cells. In addition, we show the subsequent motion of cells when induced dipoles are applied on both sides of the nucleus, thereby mimicking the two phase-shifted dipoles that we report in the text and in the model. These data further support our general mechanism.

In summary, the theoretical concept is very appealing and the experiments conducted are creative and interesting, but the relation between theory and experiment is not entirely convincing. Therefore, impact in the wider community of quantitative life sciences might remain limited for the current form of this manuscript.

We thank the referee for the positive evaluation. As we detail, we provided more experimental data and analysis to substantiate the relation between theory and experiments.

In order to improve the relation between the general concept and the supporting experimental data, it would be very informative to conduct these experiment with different matrices and different cells types (ideally even with ghosts without nucleus), to see how general this observation is. How much does pore size of CDM matters, both for the oscillations and the laser ablation experiments? Is it important that the nucleus separates these two regions of the cell? Why cannot the same experiment be one in two dimensions, where the nucleus is out of the way? It would also be interesting to consider if less perturbative experiments might be feasible (e.g. optogenetics to locally induce contractility).

We performed and now report some of the experiments suggested by the reviewer – which involved experiments within the plan of action. We had performed experiments with collagen and these conditions did not significantly modify the motion of cells, which is consistent with the report of Ben Fabry et al., Nature Methods 2015 now quoted. In addition, and as suggested by the referee, we added new experiments with two other cell types: one fibroblast cell line and primary fibroblasts. These two cell lines also display similar patterns of contraction and extension at the front and the back of the nucleus. These results are added in a new supplementary figure (Figure 1 —figure supplement 3), video and in the Results section (p. 3). In addition, we now show that cytoplasts (cell fragments without nuclei or ‘ghosts without nucleus’ asked by the reviewer) perform oscillatory motions similar to the ones observed for nocodazole treated cells. These new experiments are shown in Figure 1 – supplementary figure 3 and in Video 10, and discussed in the Results section (p. 6). Altogether, our results suggest that the oscillatory pattern is generic and does not necessarily require the nucleus.

For the theory part, it would be interesting to give more background on the calculations behind Equation 2 and to explain if it also generates an open loop in the D-Q-plane as shown here for the experimental data.

We now give more background on the calculations on p. 7. We also show the trajectories in the D-Q plane generated by the theoretical model in the new Figure 4 —figure supplement 2 and refer to this figure on p. 7.

Finally, I note that the discussions of the scallop theorem for swimming, although interesting, are not directly relevant to this work, because as the authors comment themselves, the situation is very different for crawling. Of course, this is an interesting analogy, but it also might be distracting or confusing to the general readers of eLife; it would be more interesting in the context of a physics journal. It is somehow odd that there are more references on the theory of swimming than on the theory of crawling. I note that there is actually a literature on the scallop theorem for crawling cells not cited here, most prominently the work by Pierre Recho, compare Recho, P., J-F. Joanny, and L. Truskinovsky. "Optimality of contraction-driven crawling." Physical Review Letters 112.21 (2014): 218101. I also miss references to earlier use of force multipoles in the cellular context (here mainly the work of Tanimoto and Sano is cited), and to traction force microscopy (including the work by Ben Fabry), which can be used to measure them.

We now better acknowledge the theoretical literature on crawling cells in the introduction. Instead of the proposed reference, we now cite and discuss in the introduction (p. 2) P. Recho, T. Putelat, and L. Truskinovsky: Contraction-Driven Cell Motility PRL 111, 108102 (2013), which seems more appropriate to the current context, together with other references (Blanch-Mercader and Casademunt, 2013, Maiuri et al., 2015). However, we would like to stress that these models focus on steady crawling, where there exists a permanent spatial asymmetry between the cell front and back leading to a cell-scale cytoskeletal flow. As such, these correspond to the classical “keratocyte type of motility” for which the scallop theorem is naturally satisfied. Such models do not reproduce our observations which rather support a model based on phase shifted contractile units, which is very similar to toy models initially developed for micro-swimmers. We therefore find it useful to introduce our model in this context.

[Editors' note: further revisions were suggested prior to acceptance, as described below.]

The manuscript has been improved but there are some remaining issues that need to be addressed, as outlined below:1) The revised submission is strongly improved over the initial submission. The authors have conducted new experiments (other matrix, other cell types, laser ablation), added more theory results (in particular the D-Q plots), improved image quality and analysis, and made the main text more accessible. In my view, however, there still remains some doubt about the general validity of these results. As the authors write themselves, cell migration in 3D matrices can also occur through coherent actin flows or with single rather than double dipoles, so the title "Temporal correlation between oscillating force dipoles drives 3D single cell migration" and the corresponding abstract still sound too general. Interestingly, there now seem to be at least two scallop theorems for crawling, one for actin flow by Recho and coworkers and one for the contractile oscillations described here. We suggest that it is erroneous to now only citie Recho PRL 2013, because it is really the PRL 2014 that discusses the scallop theorem.

We thank the reviewer for acknowledging the improvements made to the paper. We certainly agree with the referee that cells may use different motility strategies, especially in 3D matrices. As the referee found the title of the article too general, we changed it for "3D single cell migration driven by temporal correlation between oscillating force dipoles".

The abstract was also modified along the lines suggested by the referee.

Regarding the reference from Recho *et al.,* the PRL 2014 does make explicit mention of the “scallop theorem” (after Equation 3), and the PRL 2013 refers to “Purcell’s theorem” (after Equation 2), so we now cite both papers p. 3 l. 6.

2) The manuscript has improved with more quantification available to assess the results and more precise explanations of how the results were obtained. The quantified data appear to support the interesting finding that there are two extension/contraction dipoles at the front and back of the cell. The dipole/quadupole cycle differences (Figure 3) also appear convincing, in agreement with the previous work of Tanimoto and Sano.Yet, there remain several further points that are not fully convincing. The first main one is the dephasing between the two dipoles which is based on a single peak in the cross correlation and appears very noisy (the statistical significance of the count statistics Figure 2k should at least be provided).

We agree with the referee that assessing the statistical significance of the differences between phase-shifts distributions for non-migrating and migrating cells will strengthen the result. We performed ANOVA test on the data and found that distributions are indeed statistically different with p = 0.0246. This analysis has been added to the figure caption (Figure 2k) as well as in the Results section p. 5 l. 35.

Second, we are uneasy with the conclusions drawn for the laser ablations (Figure 5) and still do not see how they support (or not) the model. For instance, repeated ablations in the front of the cell would not lead to cell translocation given the model advocated by the authors (if we understand correctly). But this control has not been performed. Moreover, given the authors' model one would expect some dependence on the relative phase of ablations in the front and the back. But this is not attempted. We are thus unsure if one can conclude something very definite from these ablation experiments.

We agree with the referees that adding the suggested control would add clarity to the conclusions from the laser ablation experiments. Therefore, we performed these experiments and results are now reported in the Results section and Figure 5 —figure supplement 1. Briefly, we performed laser ablation repeatedly at the front of a polarized cell using two different time lags (5 minutes and 10 minutes). In both cases (repeated cuts at the front every 5 minutes and repeated cuts at the front every 10 minutes), net motion was non-existent or smaller than the one observed when cuts were performed by alternating back and front of the cell. The results are shown in Figure 5 —figure supplement 1. This control indeed strengthens our result about the coupling between front and back dipoles for directed motion. The text is modified accordingly p. 10 l. 10. As for the systematic changes of phases of ablation in the front and the back, this requires a dedicated new study in its own right and we leave study the complete characterizations for another article.

3) Overall, the reviewers are of the option that the study is strongly theory-driven and that the authors could do better in describing the experimental evidence. The whole design of the manuscript is such that the idea is put first and then some experimental evidence is provided. An alternative approach would be to start with the general status of the field of cell migration, then introduce the experimental results for this specific and certainly interesting system, and then the theory explanation as one way to explain these observations. Please consider this possible revision.

We agree with the suggestions and we modified the article accordingly. We now start the article by general status of cell migration and the strategy for the design of the experiments before outlining the need for physical approaches to explain how cell move with zero force. We also discuss that this theory is a potential way of explaining the dynamics and measurements in the discussion with a dedicated new paragraph.